# N6-methyladenosine RNA modification regulates photosynthesis during photodamage in plants

Man Zhang [1,2,3], Yunping Zeng[1], Rong Peng[1], Jie Dong[2], Yelin Lan[2], Sujuan Duan[1], Zhenyi Chang[1], Jian Ren[2], Guanzheng Luo [2], Bing Liu[2], Kamil Růžička[4], Kewei Zhao[5], Hong-Bin Wang [1,6,7] ✉ & Hong-Lei Jin [1,5] ✉

N6-methyladenosine (m6A) modification of mRNAs affects many biological processes. However, the function of m6A in plant photosynthesis remains unknown. Here, we demonstrate that m6A modification is crucial for photosynthesis during photodamage caused by high light stress in plants. The m6A modification levels of numerous photosynthesis-related transcripts are changed after high light stress. We determine that the Arabidopsis m6A writer VIRILIZER (VIR) positively regulates photosynthesis, as its genetic inactivation drastically lowers photosynthetic activity and photosystem protein abundance under high light conditions. The m6A levels of numerous photosynthesis-related transcripts decrease in *vir* mutants, extensively reducing their transcript and translation levels, as revealed by multi-omics analyses. We demonstrate that VIR associates with the transcripts of genes encoding proteins with functions related to photoprotection (such as *HHL1*, *MPH1*, and *STN8*) and their regulatory proteins (such as regulators of transcript stability and translation), promoting their m6A modification and maintaining their stability and translation efficiency. This study thus reveals an important mechanism for m6A-dependent maintenance of photosynthetic efficiency in plants under high light stress conditions.

Plants use light energy for photosynthesis, convert inorganic matter ($CO_2$ and water) into organic matter, and release oxygen, forming the basis for life on Earth. The photosynthetic apparatus on the chloroplast thylakoid membrane includes photosystem II (PSII), cytochrome $b_6f$, photosystem I (PSI), and ATP synthase, which cooperate to harvest light, transfer electrons, and convert light into chemical energy[1]. PSII and PSI harvest light energy and convert it to electron energy, which is essential for photosynthesis[2]. However, excess light can damage photosystems, especially PSII. Since PSII drives the oxidation of water and the reduction of plastoquinone and is the sensitive

[1]Institute of Medical Plant Physiology and Ecology, School of Pharmaceutical Sciences, Guangzhou University of Chinese Medicine, 510006 Guangzhou, People's Republic of China. [2]School of Life Sciences, Sun Yat-sen University, 510275 Guangzhou, People's Republic of China. [3]Institution of Fruit Tree Research, Guangdong Academy of Agricultural Sciences; Key Laboratory of South Subtropical Fruit Biology and Genetic Resource Utilization, Guangdong Provincial Key Laboratory of Tropical and Subtropical Fruit Tree Research, 510640 Guangzhou, People's Republic of China. [4]Laboratory of Hormonal Regulations in Plants, Institute of Experimental Botany, Czech Academy of Sciences, 165 02 Prague 6, Czech Republic. [5]Guangzhou Key Laboratory of Chinese Medicine Research on Prevention and Treatment of Osteoporosis, The Third Affiliated Hospital of Guangzhou University of Chinese Medicine, No.263, Longxi Avenue, Guangzhou, People's Republic of China. [6]Key Laboratory of Chinese Medicinal Resource from Lingnan (Guangzhou University of Chinese Medicine), Ministry of Education, Guangzhou, People's Republic of China. [7]State Key Laboratory of Dampness Syndrome of Chinese Medicine, Guangzhou University of Chinese Medicine, Guangzhou, People's Republic of China. ✉e-mail: wanghongbin@gzucm.edu.cn; jinhl@gzucm.edu.cn

photochemical reaction center, PSII is particularly susceptible to photodamage or photoinhibition under high light conditions, resulting in lower photosynthetic efficiency and impairing growth and development[2,3].

To maintain photosynthetic activity under high light conditions, plants have evolved sophisticated photoprotective mechanisms, including leaf and chloroplast movements[4,5], screening against damaging radiation[6], state transitions, adjusting the PSII/PSI ratio and the size of the light-harvesting antennae[7–9], scavenging of reactive oxygen species[10], energy-dependent non-photochemical quenching (NPQ)[11], cyclic electron flow around PSI, and photorespiration[12]. PSII repair is especially important for efficient photosynthetic activity under photo-damaging conditions[13]. The regulation of gene expression is an important aspect of plant responses and acclimation to light stress[14,15]. The abundance of photosynthesis proteins is altered to respond to light stress[16–19]. Notably, the expression of numerous genes encoding proteins with functions related to photoprotection (photoprotection-related genes) is activated to protect the photosystem apparatus from high light conditions[20]. In particular, regulating the expression of factors related to PSII photoprotection is also critical for maintaining photosynthetic efficiency and protecting PSII under high light conditions[21].

N[6]-methyladenosine (m[6]A), the most prevalent internal post-transcriptional modification in the messenger RNAs (mRNAs) of most eukaryotes, is widely distributed in mammals, plants, viruses, yeast (*Saccharomyces cerevisiae*), flies, and bacteria[22–25]. This modification plays broad roles in regulating RNA metabolism, including the splicing, nuclear export, stability, localization, and translation of mRNAs[26–29]. m[6]A is dynamically written, erased, and read via a complex network of m[6]A writer, eraser, and reader proteins[23,25,29,30]. In plants, m[6]A methylation is catalyzed by a methyltransferase complex (writer) containing mRNA ADENOSINE METHYLASE (MTA), METHYLTRANSFERASE B (MTB), FKBP12 INTERACTING PROTEIN37 (FIP37), VIRILIZER (VIR), and HAKAI[31–34]. The m[6]A modification is reversible and can be removed by the m[6]A demethylases (erasers) ALKBH9B and ALKBH10B[35,36]. The m[6]A-modified RNAs are recognized by reader proteins such as EVO-LUTIONARILY CONSERVED C-TERMINAL REGION2 (ECT2), ECT3, ECT4, and CLEAVAGE AND POLYADENYLATION SPECIFICITY FAC-TOR30 (CPSF30)[37–40], which help implement the biological functions of the m[6]A modification.

The m[6]A modification plays a critical role in plant embryonic development. Arabidopsis (*Arabidopsis thaliana*) mutants lacking MTA, MTB, FIP37, or VIR function are all embryo-lethal[31,41]. Partial loss of function of the m[6]A writer also leads to many additional plant phenotypes, such as reduced apical dominance, reduced root growth, aberrant gravitropic responses, trichomes with more branches, and the overproliferation of shoot meristems[29,30]. m[6]A erasers and readers also modulate various biological processes, as their mutants exhibit distinct phenotypes. The Arabidopsis *alkbh10b* mutant shows delayed flowering and repressed vegetative growth[36]. The genetic inactivation of *ALKBH9B* led to a lower incidence of viral infections[35]. ECT proteins, including ECT2 to ECT4, control developmental timing and morphogenesis in Arabidopsis[37,38,42]. Therefore, the m[6]A modification regulates a variety of biological processes in plants. Interestingly, previous studies suggested that m[6]A modifications may involve chloroplast/photosynthetic genes[24,43]. However, the role of the m[6]A modification in photosynthesis remains largely unclear.

In the current study, we analyzed changes in m[6]A modification of transcripts in response to high light stress and established that m[6]A modification levels of numerous photosynthesis-related transcripts are altered after high light stress. We demonstrated that VIR, a member of the m[6]A writer complex, plays a vital role in maintenance of photosynthetic efficiency under high light conditions. The partial loss of VIR function caused dramatically reduced photosynthetic efficiency and defects in m[6]A modification of numerous photosynthesis-related transcripts under high light conditions. Moreover, we reveal the

mechanism by which VIR regulates the m[6]A modification of photoprotection-related transcripts, thereby regulating the expression of these genes in multiple post-transcriptional processes.

## Results

### High light induces significant changes in m[6]A modification of transcripts for chloroplast/photosynthetic genes

To test whether the m[6]A modification is linked with high light stress, we first compared the total m[6]A levels of mRNA in Arabidopsis Col-0 seedlings before and after a 4-h and 12-h high-light (HL) treatment by dot blot analysis using m[6]A antibody. We observed that total m[6]A levels are upregulated in Col-0 after HL treatment (Fig. 1a). Quantitative measurement of m[6]A levels by liquid chromatography-tandem mass spectrometry (LC-MS/MS) revealed that total m[6]A levels of mRNA from Arabidopsis seedlings after the 4-h and 12-h HL treatment increase by 15% and 21% compared to that before HL treatment (Fig. 1b), supporting the idea that HL induces an increase of overall m[6]A levels of mRNA.

Next, we performed m[6]A RNA immunoprecipitation and sequencing (m[6]A-seq)[44] on poly(A) RNA from Arabidopsis Col-0 seedlings before (Col-0_0h HL) and after (Col-0_4h HL) a 4-h HL treatment. We identified m[6]A peaks by comparing the read counts between m[6]A-immunoprecipitation (IP) and input data in the continuous region of each transcript using the R package exomePeak with default parameters and a *P*-value < 0.05. We detected 18,809–20,451 m[6]A peaks for each biological replicate and selected peaks present in both replicates as "confident peaks" for subsequent analysis (Fig. 1c). We thus identified 14,249 confident m[6]A peaks representing the transcripts of 12,481 genes in Col-0_0h, and 14,150 m[6]A peaks representing the transcripts of 12,448 genes in Col-0_4h (Fig. 1c, Supplementary Data 1 and 2). The two biological replicates of each set of samples had high Pearson's correlation coefficients (Supplementary Fig. 1a). We validated the m[6]A-seq results via independent m[6]A-immunoprecipitation (IP)-qPCR of m[6]A-containing transcripts and negative control transcripts reported in a previous study[24]. m[6]A-containing transcripts were significantly enriched in the IP samples, while we detected almost no enrichment for the negative control transcripts (Supplementary Fig. 1b, c). These results suggested that our m[6]A-seq data are accurate and robust. In addition, many m[6]A peaks contained the m[6]A consensus motif RRACH (R = A/G, H = A/C/U) or the plant-specific m[6]A motif URUAY (R = G > A, Y = U > A) (Supplementary Fig. 1d). We used the R package exomePeak to identify differentially methylated m[6]A peaks (with a false discovery rate [FDR] < 0.05 and fold-change ≥ 2) in Col-0 seedlings before and after HL treatment. Compared to Col-0_0h, we identified 312 significantly downregulated m[6]A peaks covering 304 genes and 99 significantly upregulated m[6]A peaks covering 94 genes in Col-0_4h (Fig. 1d, Supplementary Data 3). The m[6]A modification levels of several chloroplast/photosynthesis-related genes changed significantly in response to HL treatment (Fig. 1e, Supplementary Table 3). For example, the m[6]A modification levels of nucleus-encoded plastid RNA polymerase *RpoTmp*[45] transcripts increased after a 4-h HL treatment, whereas the m[6]A modification levels of *PHYTOCHROME KINASE SUB-STRATE1* (*PKS1*)[46] transcripts decreased under the same conditions (Fig. 1f). These results suggest that the m[6]A modification levels of chloroplast/photosynthetic genes are regulated by HL.

We further analyzed the gene expression pattern of m[6]A regulators in response to HL stress. Accordingly, we measured the gene expression pattern of m[6]A regulator genes after different times under HL treatment, from 0 to 12 h. Notably, the mRNA levels of genes encoding m[6]A writers (*MTA*, *MTB*, *FIP37*, *VIR*, and *HAKAI*) and m[6]A readers (*ECT2*, *ECT3*, and *ECT4*) were substantially induced by HL, and the m[6]A eraser gene (*ALKBH10B*) was moderately upregulated after this treatment (Fig. 1g, h). To rule out possible effects from different treatment durations, we also compared the expression levels of m[6]A regulator genes in seedlings grown under different HL treatment for different times and corresponding normal growth light conditions:

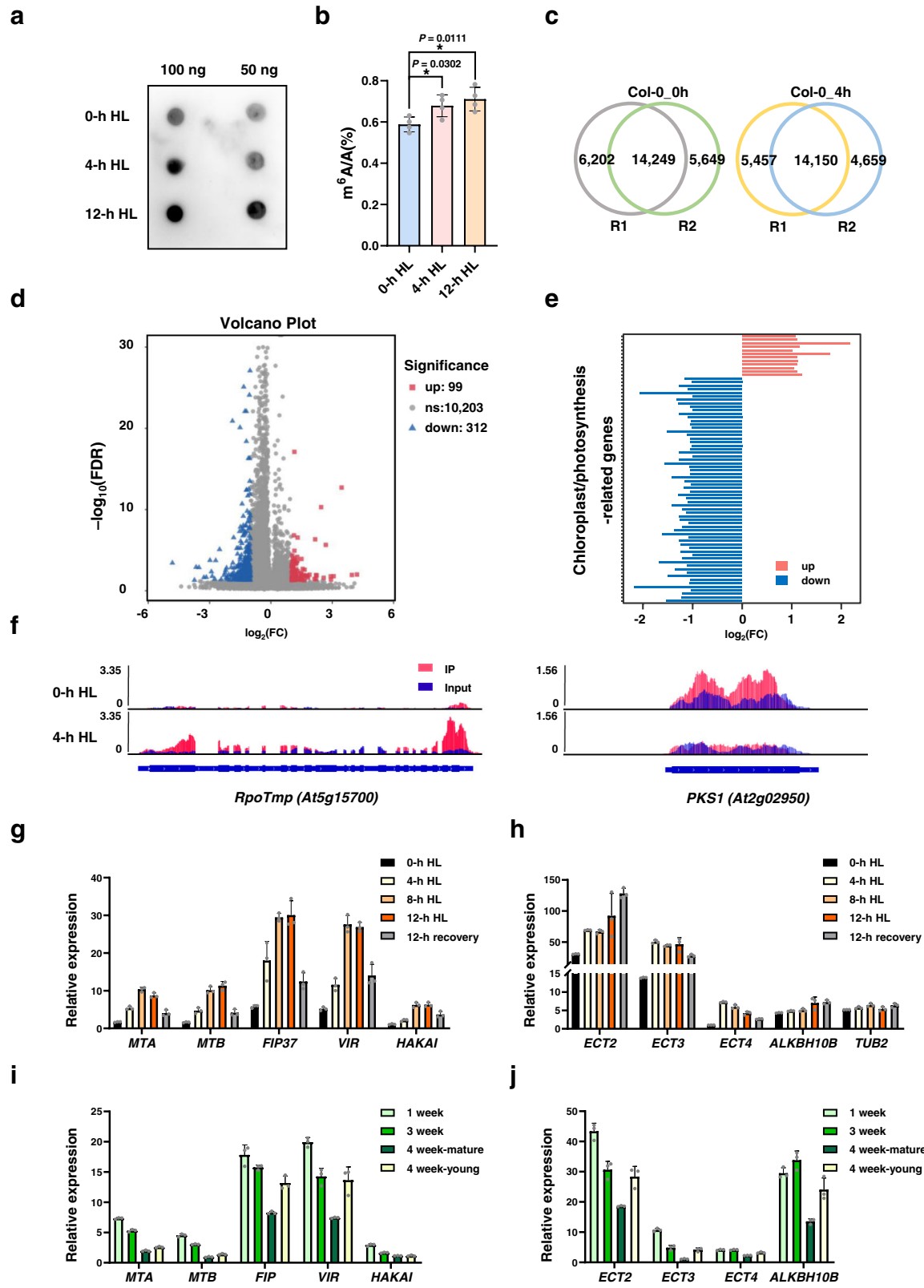

The expression of m⁶A regulator genes, especially that of m⁶A writer genes, was drastically induced after HL treatment compared to normal growth light conditions (Supplementary Fig. 2), suggesting that HL-induced gene expression is not due to diurnal or circadian effects. These results are consistent with the observed upregulation of the overall m⁶A level in wild-type Arabidopsis seedlings (Fig. 1a, b). Additionally, the expression levels of genes encoding m⁶A regulators were

highest in 1-week-old seedlings and gradually decreased over time (Fig. 1i, j).

## The *vir-1* mutants are hypersensitive to high light stress

To explore the role of m⁶A modification in plant photoprotection and photosynthesis, we systematically screened 1-week-old Arabidopsis m⁶A regulator mutants using a chlorophyll fluorescence video imaging

**Fig. 1 | Effects of high light on the m⁶A modification in plants. a** Dot blot analysis of m⁶A levels in mRNA purified from Col-0 seedlings before and after 4 h or 12 h of high light (HL) treatment. **b** m⁶A percentage relative to adenosine (m⁶A/A %) determined by LC-MS/MS in mRNA purified from Col-0 seedlings before and after 4 h or 12 h of HL treatment. Values are means ± standard error (SE) ($n = 4$ biological replicates). *$P < 0.05$; **$P < 0.01$, by two-sided Student's $t$ test. **c** Overlapping m⁶A peaks identified in biological replicates (R1-R2) of control Col-0 seedlings (Col-0_0 h HL) or exposed to high light (HL) for 4 h (Col-0_4 h HL). **d** Volcano plot of the significant differences in m⁶A peaks between Col-0_0 h HL and Col-0_4 h HL seedlings across two biological replicates. **e** Changes in m⁶A modification levels of chloroplast/photosynthesis-related genes after 4-h HL treatment. Corresponding

gene IDs are listed in Supplementary Table 3. **f** Integrative Genomics Viewer windows displaying examples of high light–mediated changes in m⁶A modification. Blue, input reads; red, IP reads. The gene models are shown below, with thick boxes and lines representing exons and introns, respectively. **g, h** qRT-PCR analysis of the expression patterns of m⁶A writer (**g**) and reader and eraser (**h**) genes under HL treatment and subsequent recovery under normal growth light conditions for 12 h. *UBQ10* was used as an internal control. Values are means ± SE ($n = 3$ biological replicates). **i, j** qRT-PCR analysis of the expression patterns of m⁶A writer (**i**) and reader and eraser (**j**) genes in plants at different growth stages. *UBQ10* was used as an internal control. Values are means ± SE ($n = 3$ biological replicates).

system (Supplementary Fig. 3 and Fig. 2a), as the genes encoding m⁶A regulators were expressed at their highest levels at this stage (Fig. 1i, j). We determined that *vir-1* mutants are most hypersensitive to HL stress. The *vir-1* mutant used here contained a G-to-A conversion mutation in the first nucleotide of intron 5 of the *VIR* gene (At3g05680), resulting in a failure of most *VIR* transcripts to be spliced correctly, resulting in a partial loss-of-function allele (Supplementary Fig. 4a), in agreement with previously reported results[34]. Compared to Col-0, the *vir-1* mutant (Supplementary Fig. 4a) had a lower maximum photochemical efficiency of PSII ($F_v/F_m$) after a 4-h HL treatment (1,000 μmol photons m⁻² s⁻¹) (Fig. 2a). The $F_v/F_m$ values of Col-0 and the *vir-1* mutants were identical when grown under normal growth light conditions (120 μmol photons m⁻² s⁻¹), although *vir-1* mutant seedlings accumulated less biomass than Col-0 (Fig. 2b). Moreover, the $F_v/F_m$ values of the *vir-1* mutant returned to normal levels following a 2-day recovery under normal growth light conditions (Fig. 2a, c). This aberrant $F_v/F_m$ phenotype after HL exposure was fully rescued by a complementation construct with the *VIR* gene (Fig. 2a, c). These results indicated that the partial loss of VIR function disrupts PSII capacity under HL conditions, suggesting that VIR is important for maintaining PSII capacity after HL exposure. In addition, the parameters qP and qL were lower in the *vir-1* mutant relative to the wild type under normal growth light conditions, pointing to their low photosynthetic activity/low utilization efficiency of light (Supplementary Table 1).

Although the *vir-1* mutant and wild-type plants were the same age, most mutant plants were less developed, and some *vir-1* mutant plants varied in size. To determine whether the photosensitive phenotype of *vir-1* is caused by delayed development, we selected mutant and wild-type seedlings from the same batch at a similar developmental stage (similar sized plants) to compare their photosynthetic activity after 4 h of HL treatment. We discovered that $F_v/F_m$ is also significantly lower in the *vir-1* mutant compared to Col-0 after a 4-h HL exposure (Supplementary Fig. 5), suggesting that the photosensitive phenotype of *vir-1* is not caused by delayed development. To confirm this result, we constructed β-estradiol-inducible RNA interference (RNAi) lines (*VIR*-RNAi-1 and *VIR*-RNAi-2), sprayed 4-day-old RNAi seedlings with β-estradiol for 3 days and then subjected them to a 4-h HL treatment. The β-estradiol-induced *VIR* RNAi lines phenocopied the *vir-1* mutants under the 4-h HL condition (Supplementary Fig. 6), which further supported the notion that the observed differences between *vir-1* and the wild type are not due to a developmental delay.

To further characterize the photosynthetic activity of PSII in the *vir-1* mutant, we analyzed the light intensity dependence of the following chlorophyll fluorescence parameters: the light-response curves of PSII quantum yield (ΦPSII), electron transport rate (ETR), and non-regulated non-photochemical quenching yield [Y(NO)] before (0 h) and after 4 h of HL treatment. Both ΦPSII and ETR parameters were much lower in the *vir-1* mutant than in Col-0 and the complemented mutant line after 4 h of HL (Fig. 2d), consistent with the $F_v/F_m$ results (Fig. 2a, c). After extending the HL treatment to 24 h, the difference between Col-0/ or the complemented mutant line and the *vir-1* mutant was much greater than that observed after 4 h of HL treatment (Fig. 2d). In addition, the sharp decrease in photosynthetic activity in

*vir-1* after a 24-h HL exposure did not result in a significant difference in cell death relative to Col-0 (Supplementary Fig. 4c). ΦPSII and ETR in the *vir-1* mutant treated with HL for 4 h almost reached wild-type levels after the plants were returned to normal growth light conditions for 2 d. Y(NO), which reflects passive energy dissipation, was much higher in the *vir-1* mutant than the wild-type controls before and (especially) after HL treatment (Fig. 2d), suggesting that the ability of the *vir-1* mutant to mount an effective protection against HL is suboptimal. Energy-dependent non-photochemical quenching (NPQ) is an important photoprotective strategy that converts excess absorbed light energy into thermal energy[11,47]. Both before and after HL, NPQ rose rapidly in Col-0, in contrast to a more gradual increase in the *vir-1* mutant, although the mutant eventually reached values comparable to those of Col-0 (Fig. 2e). This phenotype might be another important characteristic of the defective photoprotective ability of the *vir-1* mutant: When HL occurs suddenly, *vir-1* cannot dissipate the excess energy as quickly as Col-0 can.

## Photosystem complex accumulation is disrupted in the *vir-1* mutant after high light treatment

The above results indicated that the photosynthetic activity of PSII is reduced in *vir-1* plants after HL treatment, suggesting that VIR is involved in photoprotection against HL stress. To investigate the effects of VIR on the structure and function of the thylakoid photosynthetic apparatus, we analyzed the abundance of various photosynthetic complexes in Col-0 and *vir-1* seedlings. Specifically, we analyzed proteins from thylakoid membranes that had been solubilized in 2% n-dodecyl-β-D-maltoside (DM) and separated by blue native PAGE (BN-PAGE) followed by immunoblotting with specific antibodies. Immunoblotting with anti-D1, anti-D2, anti-CP43, and anti-CP47 antisera revealed that thylakoid membranes from *vir-1* seedlings contain fewer PSII complexes than those from Col-0 seedlings after HL treatment (Fig. 3a, b). In addition, under both normal and HL conditions, the levels of PSI and Cyt $b_6f$ complexes were lower in *vir-1* than in Col-0, while the levels of ATPase complexes were similar between the two genotypes (Supplementary Fig. 4d, e). These results suggested that lower functional VIR levels affect the accumulation of PSII complexes upon HL treatment.

To explore whether the lower abundance of photosystem complexes was related to changes in the levels of related subunits, we performed immunoblot analysis using antibodies against the subunits of the thylakoid membrane photosynthetic protein complexes. After HL treatment, we observed marked reductions in the levels of the PSII core subunits D1, D2, CP43, and CP47 in the *vir-1* mutant, reaching about 57%, 79%, 58%, and 42% of Col-0 levels, respectively (Fig. 3c, d). The cytochrome $f$ levels were much lower in the mutant under both normal growth light and HL conditions relative to Col-0 (Fig. 3c, d). By contrast, the levels of PsbO (another PSII protein), the light-harvesting complex II (LHCII) chlorophyll $a/b$ binding proteins LHCA1 and LHCB1, and ATP synthase subunit B accumulated to comparable levels in *vir-1* and Col-0 plants under both light conditions (Fig. 3c, d). These results suggested that the functional deficiency of VIR perturbs the accumulation of PSII core subunits in Arabidopsis after HL treatment.

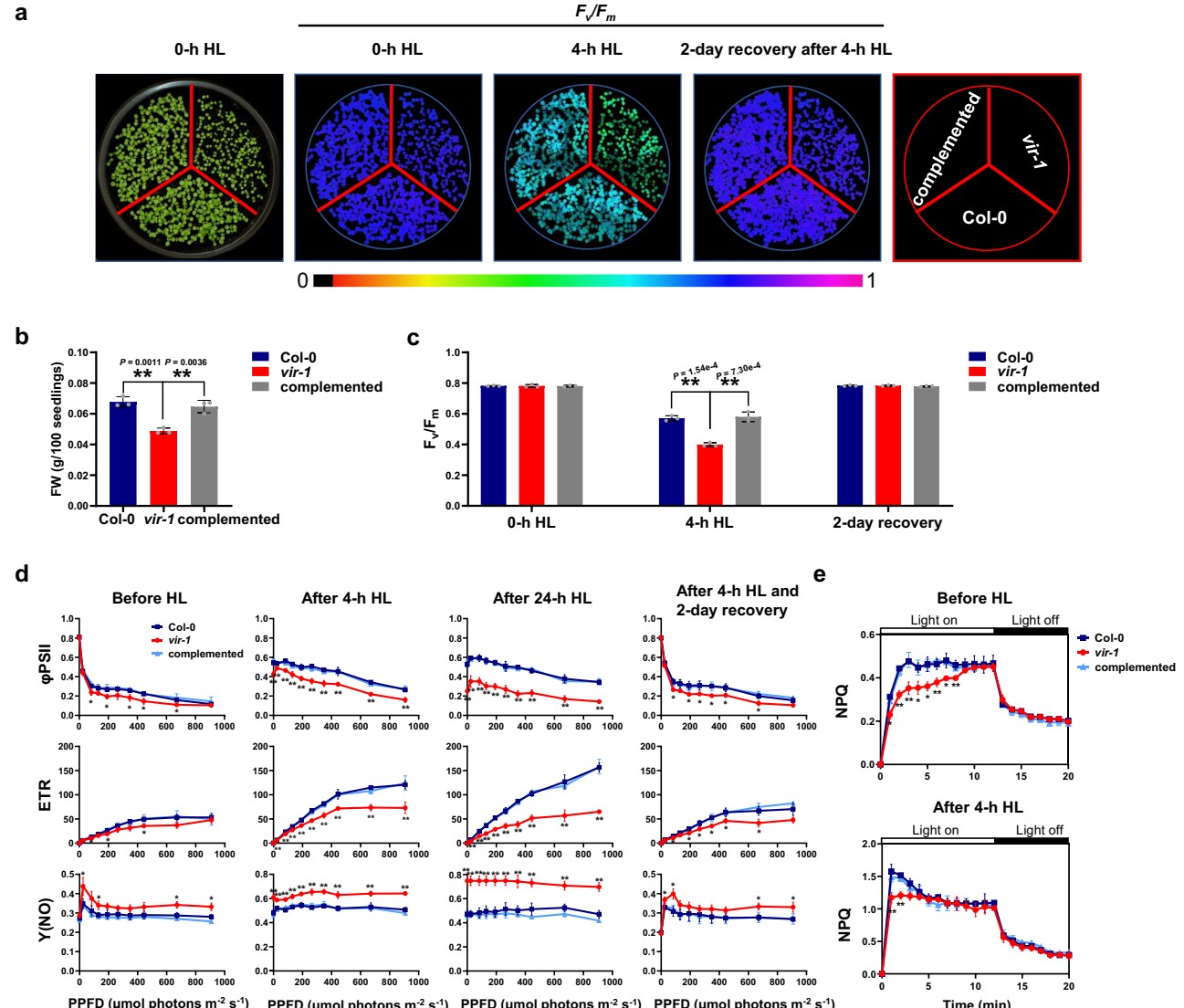

**Fig. 2 | Characterization of photosynthetic activity in Col-0 and *vir-1* seedlings.**
**a** Left panel, image of 7-day-old Col-0, *vir-1*, and *VIR*-complemented seedlings under normal growth light. Center panels, fluorescence images used to measure $F_v/F_m$ of seedlings grown under different light conditions. False-color images representing $F_v/F_m$ after 0 or 4-h high light (HL) treatment and 2-day recovery after 4-h HL treatment in Col-0, *vir-1*, and complemented seedlings are shown. The false color ranges from black (0) to purple (1), as indicated in the scale at the bottom. **b** Fresh weight (FW) of 7-day-old Col-0, *vir-1*, and complemented seedlings. Values are means ± SE (*n* = 3 biological replicates). *$P < 0.05$; **$P < 0.01$, by two-sided Student's *t* test. **c**, Changes in $F_v/F_m$ values for 7-day-old Col-0, *vir-1*, and complemented seedlings under HL treatment. Values are means ± SE (*n* = 3 biological replicates). *$P < 0.05$; **$P < 0.01$, by two-sided Student's *t* test. **d** Light-response curves of PSII quantum yield (ΦPSII), electron-transport rate (ETR), and non-regulated non-photochemical quenching yield [Y(NO)] in Col-0, *vir-1*, and complemented seedlings after a 0-, 4-, or 24-h HL treatment and 2-day recovery after a 4-h HL treatment. Measurements were performed under the following light intensities: 0, 24, 83, 130, 192, 264, 348, 444, 671, and 908 µmol photons m$^{-2}$ s$^{-1}$. PPFD, Photosynthetic photon flux density. Values are means ± SE (*n* = 3 biological replicates). Asterisks indicate a significant difference between Col-0 and *vir-1* using the two-sided Student's *t* test. *$P < 0.05$; **$P < 0.01$. No significant differences were detected between Col-0 and complemented seedlings. **e** Time courses for the induction and relaxation of NPQ before and after 4-h HL. Actinic light (500 µmol photons m$^{-2}$ s$^{-1}$) was switched on at time zero, and seedlings were left in the dark after 12 min. Values are means ± SE (*n* = 3 biological replicates). Asterisks indicate a significant difference between Col-0 and *vir-1* using the two-sided Student's *t* test. *$P < 0.05$; **$P < 0.01$. No significant differences were detected between Col-0 and complemented seedlings.

To investigate the effects of VIR on PSII protein stability, we blocked the biosynthesis of chloroplast-encoded proteins in *vir-1* and Col-0 seedlings using chloramphenicol and measured PSII protein abundance by immunoblot analysis. In the absence of chloramphenicol, the levels of PSII proteins D1, D2, CP43, and CP47 declined gradually in both Col-0 and *vir-1* seedlings; the extent of the decrease was higher in *vir-1* seedlings (especially D1) after a 4-h HL treatment. In the presence of chloramphenicol, the levels of PSII core subunits D1, D2, CP43, and CP47 declined rapidly in both Col-0 and *vir-1* seedlings with similar degradation rates (Supplementary Fig. 7).

These results suggested that the low levels of PSII subunits in *vir-1* seedlings upon HL exposure are not due to the rapid degradation of these proteins.

### The mRNA m⁶A modification of photoprotection-related genes is defective in the *vir-1* mutant

To test the global effects of VIR-mediated m⁶A modification and investigate how loss of VIR function results in the HL hypersensitivity phenotype of the *vir-1* mutant, we generated transcriptome-wide m⁶A profiles of Col-0 and *vir-1* seedlings before and after HL treatment by

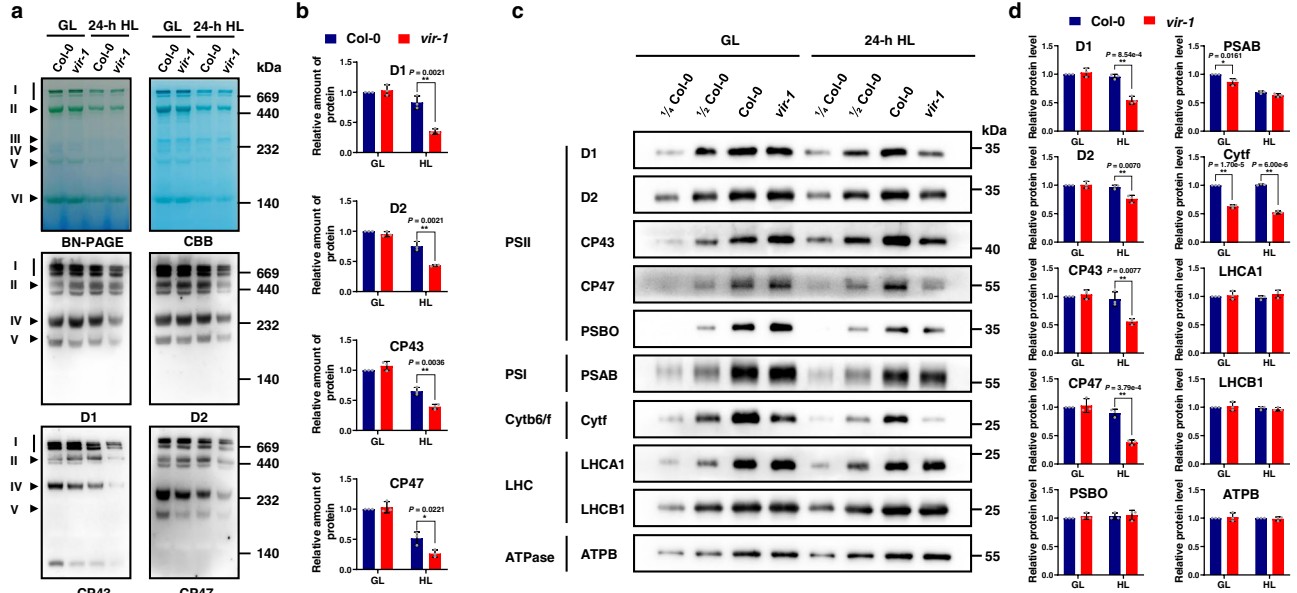

**Fig. 3 | Analysis of PSII complexes and subunits in Col-0 and *vir-1* seedlings. a** BN-PAGE and immunoblot analysis of thylakoid photosynthetic complexes. Equal amounts of thylakoid membrane (10 µg chlorophylls) from Col-0 and *vir-1* seedlings under 0-h (GL) and 24-h (HL) high light treatment were solubilized with 2% n-dodecyl-β-D-maltoside (DM) and separated by BN-PAGE. The BN-PAGE gel was stained with Coomassie brilliant blue (CBB). The macromolecular protein complexes of thylakoid membranes (indicated on the left) were identified according to Jin[48]. For BN-PAGE immunoblot analysis, an equal amount of chlorophyll (1.5 µg) was loaded in each lane, and anti-D1, anti-D2, anti-CP43, and anti-CP47 antisera were used to probe the PSII complex. I: PSII-LHCII supercomplex; II: PSII dimer, PSI monomer; III: PSI monomer, CF1; IV: Cyt $b_6f$, PSII core monomer; V: CP43-less PSII core monomer; VI: LHCII trimer. All experiments involved three independent biological replicates, which produced similar results. **b** Proteins immunodetected

from (**a**) were quantified with Phoretix 1D software (Phoretix International, UK). Values (mean ± SE, $n = 3$ independent biological replicates) are given relative to protein levels of Col-0 before HL treatment. *$P < 0.05$; **$P < 0.01$, by two-sided Student's *t* test. **c**, Analysis of *t*hylakoid membrane protein accumulation in Col-0 and *vir-1* mutants. Thylakoid membrane proteins from Col-0 and *vir-1* seedlings were separated by 12% SDS-urea-PAGE and probed with antisera against specific thylakoid membrane proteins. Samples were loaded on an equal chlorophyll basis. Cyt $b_6f$, cytochrome $b_6f$ complex; LHC, light-harvesting complex; ATPase, ATP synthase complex. Similar results were obtained from three independent biological replicates. **d** Proteins immunodetected from (**c**) were quantified with Phoretix 1D software (Phoretix International, UK). Values (means ± SE; $n = 3$ independent biological replicates) are given relative to protein levels of Col-0 before high light treatment. *$P < 0.05$; **$P < 0.01$, by two-sided Student's *t* test.

m⁶A-seq. The Pearson's correlation coefficients were high between the biological replicates for each set of samples (Supplementary Fig. 8a). We detected 20,136–21,061 m⁶A peaks in each biological replicate for *vir-1* seedlings (Supplementary Fig. 8b and Supplementary Data 4 and 5). We evaluated the distribution of m⁶A peaks in Col-0 and *vir-1* seedlings. The m⁶A peaks of Col-0 were greatly enriched near the stop codon (52.0–52.2%), start codon (10.7–12.9%), and 3' untranslated region (UTR, 4.3%–5.4%), which was also consistent with the distribution of m⁶A modifications previously identified in Col-0[36]. By contrast, the m⁶A peaks in *vir-1* seedlings showed an enrichment in coding sequences (CDSs) (43.9–46.5%), while the enrichment near the stop codon (26.3–27.2%) and 3' UTR (3.5–4.2%) was largely abolished (Fig. 4a, b and Supplementary Fig. 8c, d). These results suggest that the topology of the m⁶A methylome in *vir-1* mutants is dramatically altered compared to that in the wild type, regardless of HL treatment. However, the overall m⁶A distribution in Col-0 and *vir-1* was not altered by the HL treatment (Fig. 4a, b and Supplementary Fig. 8c, d).

To identify VIR-dependent m⁶A sites that might regulate photoprotection, we compared the methylation levels of each m⁶A site between Col-0 and *vir-1* seedlings before and after HL treatment using the R package exomePeak to identify differentially methylated m⁶A peaks (FDR < 0.05 and fold-change ≥ 2). We identified 3994 and 3705 VIR-dependent m⁶A peaks in Col-0 before and after HL exposure, respectively (Supplementary Data 6 and 7). The methylation levels of these m⁶A peaks were much lower or completely abolished in *vir-1* compared to Col-0. Kyoto Encyclopedia of Genes and Genomes (KEGG) analysis revealed that the genes within which these differential m⁶A peaks lie primarily function in several basic cellular metabolic processes, such as ubiquitin-mediated proteolysis, nucleotide excision

repair, the tricarboxylic acid cycle (TCA cycle), RNA transport, RNA degradation, mRNA surveillance, protein export, and oxidative phosphorylation. Notably, the ribosome category was significantly enriched and had the most genes with differential m⁶A peaks (Fig. 4c and Supplementary Fig. 8e). Moreover, Gene Ontology (GO) analysis of these genes revealed that the chloroplast category is also significantly enriched (Fig. 4d and Supplementary Fig. 8f). These results suggested that VIR-dependent m⁶A modification might play important roles in chloroplast function and photosynthesis.

Further analysis indicated that the m⁶A modification levels of many photoprotection-related transcripts are much lower in *vir-1* (Supplementary Fig. 9), including in *HYPERSENSITIVE TO HIGH LIGHT1* (*HHL1*)[48], *Maintenance of PSII under High light 1* (*MPH1*)[49], *HIGH CHLOROPHYLL FLUORESCENCE244* (*HCF244*)[50], *Degradation-of-periplasmic-proteins proteases 1* (*Deg1*)[51], *chloroplast twin-arginine translocation protein C* (*cpTatC*)[52], and *STATE TRANSITION8* (*STN8*)[53] (Fig. 4e and Supplementary Fig. 8g). Notably, we detected significant m⁶A peaks within the stop codon regions of the mRNAs of *HHL1*, *MPH1*, *HCF244*, *Deg1*, *cpTatC*, and *STN8* in Col-0 seedlings, but the peaks were much lower in the *vir-1* mutant (Fig. 4e and Supplementary Fig. 8g). We confirmed these changes in m⁶A peaks by independent m⁶A-IP-qPCR using fragmented poly(A) RNA (Supplementary Fig. 8h).

## VIR-dependent m⁶A modification mediates post-transcriptional regulation of photoprotection-related genes

To test the effect of VIR on gene expression, we subjected Col-0 and *vir-1* seedlings exposed to HL for 0 or 4 h to transcriptome deep sequencing (RNA-seq) and Ribosome profiling (Ribo-seq). We thus obtained datasets for the transcriptome and translatome (Supplementary Fig. 11a–f), both

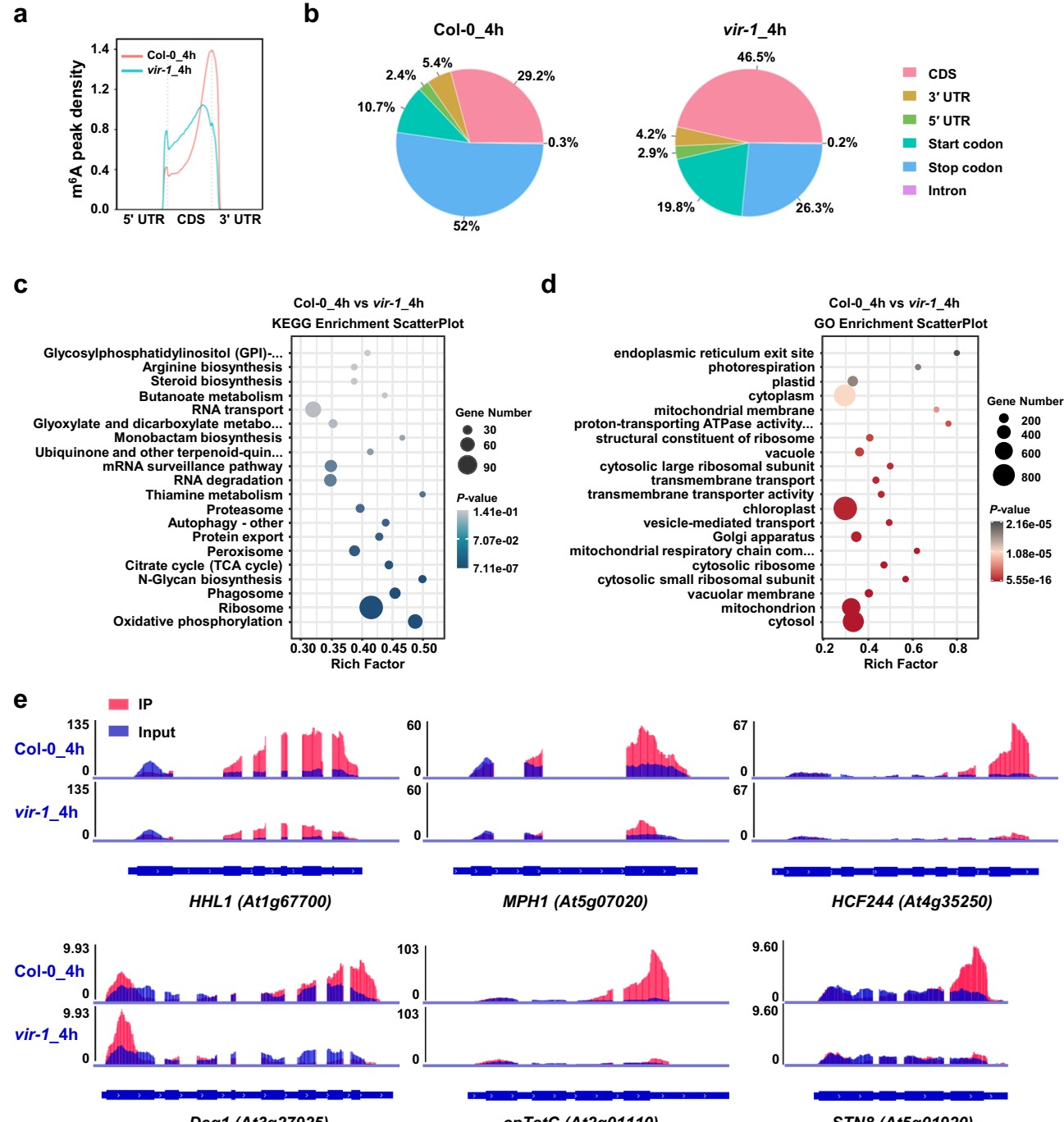

**Fig. 4 | Comparison of m⁶A modification between Col-0 and *vir-1* seedlings after high light treatment. a** Comparison of the distribution of m⁶A peaks along transcript segments in Col-0 and *vir-1* seedlings after a 4-h high light (HL) treatment. **b** Comparison of the distribution of m⁶A peaks in different segments of Col-0 and *vir-1* transcripts after a 4-h HL treatment. CDS, coding sequence; UTR, untranslated region. **c** KEGG analysis of VIR-dependent m⁶A-containing genes after a 4-h HL treatment. Statistical test was determined by one-sided hypergeometric test. **d** GO analysis of VIR-dependent m⁶A-containing genes after a 4-h HL treatment. Statistical test was determined by one-sided hypergeometric test. **e** Loss of VIR function results in lower (or loss of) m⁶A peaks compared to Col-0 seedlings after a 4-h HL. Blue, input reads; red, IP reads. The gene models are shown below, with thick boxes and lines representing exons and introns, respectively.

with two highly reproducible biological replicates (Supplementary Fig. 11g, h). The Ribo-seq data illustrated the following characteristics, including the abrupt appearance of a footprint signal 13–14 nucleotides (nt) upstream of the start codon, a rapid decay of the signal around 13-14 nt upstream of the stop codon, low footprint density in the 3′ and 5′ UTRs, and a strong 3-nt periodicity (Supplementary Fig. 11c–f), which we did not observe in the RNA-seq libraries (Supplementary Fig. 11c–f), pointing to the high quality of the Ribo-seq data[54].

Compared to Col-0, we identified 934 upregulated and 453 downregulated genes in *vir-1* before HL treatment based on analysis of differences in the transcriptome (FDR < 0.05 and fold-change > 2) (Supplementary Fig. 12a, g, h and Supplementary Data 8). GO enrichment analysis showed that these genes are mainly enriched in processes related to cell growth and development, including cell division and the mitotic cell cycle (Supplementary Fig. 12i). Upon exposure to 4 h of HL, we identified 817 upregulated and 594 downregulated genes

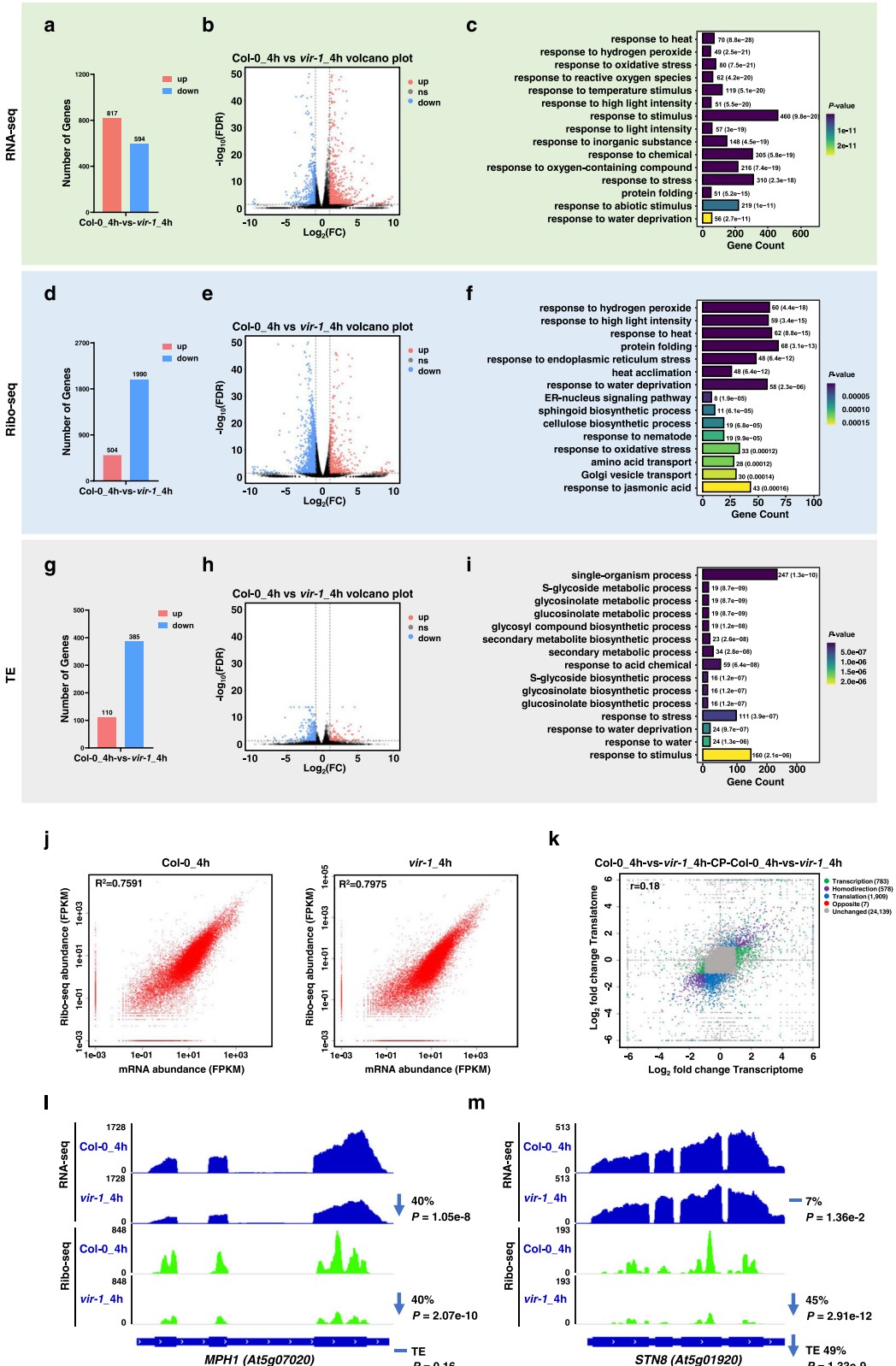

in *vir-1* compared to Col-0 (Fig. 5a, b, Supplementary Fig. 12b, and Supplementary Data 9). These genes were primarily enriched in GO terms involved in stress responses and related to the HL response, such as response to oxidative stress, response to reactive oxygen species, and response to HL intensity (Fig. 5c). Notably, the mRNA levels of photoprotection-related genes were significantly lower in *vir-1* compared to Col-0 (Fig. 6c and Supplementary Fig. 13e).

The human RNA demethylase FTO (fat mass and obesity associated) induces transcriptional activation in plants, suggesting a role for m6A in transcription[55]. Thus, we tested the effect of VIR on overall transcription. First, we performed quantitative RNA-seq with an External RNA Controls Consortium (ERCC) RNA spike-in control. The *vir-1* mutant seedlings accumulated comparable overall levels of poly(A) RNA compared to wild-type seedlings (Supplementary

**Fig. 5 | Transcriptome and translatome analysis of Col-0 and *vir-1* seedlings.**
**a** Number of differentially expressed genes between Col-0 and *vir-1* after a 4-h high light (HL) treatment. **b** Volcano plot of differentially expressed genes between Col-0 and *vir-1* after a 4-h HL treatment. Significantly downregulated genes are shown in blue, significantly upregulated genes are shown in red, and genes without significant differences in expression are shown in black. Black vertical lines highlight Log$_2$(fold-change) = 1 or −1; black horizontal line represents FDR of 0.05. **c** GO enrichment analysis of the differentially expressed genes between Col-0 and *vir-1* after a 4-h HL treatment. Statistical test was determined by one-sided hypergeometric test. **d** Number of differentially translated genes between Col-0 and *vir-1* after a 4-h HL treatment. **e** Volcano plot of differentially translated genes between Col-0 and *vir-1* after a 4-h HL treatment, indicated as in (**b**). **f** GO enrichment analysis of the differentially translated genes between Col-0 and *vir-1* after a 4-h HL treatment. Statistical test was determined by one-sided hypergeometric test. **g** Number

of genes showing differential translation efficiency (TE) between Col-0 and *vir-1* after a 4-h HL treatment. **h** Volcano plot of differential TE genes between Col-0 versus *vir-1* after a 4-h HL treatment, indicated as in (**b**). **i** GO enrichment analysis of the differential TE genes between Col-0 and *vir-1* after a 4-h HL treatment. Statistical test was determined by one-sided hypergeometric test. **j** Extent of correlation between RNA-seq and Ribo-seq samples after a 4-h HL treatment. Pearson's correlation coefficient (*r*) is shown. **k** Extent of correlation between the transcriptome changes and translatome changes in *vir-1 vs*. Col-0 after a 4-h HL treatment. Pearson's correlation coefficient (*r*) is shown. **l**, **m** Normalized distribution of RNA-seq and Ribo-seq reads in Col-0 and *vir-1* along the *MPH1* (**l**) and *STN8* (**m**) genes. The fold-change and associated *P*-value for VIR effect on transcript and footprint levels, as well as the fold-change in the footprint levels given the levels of mRNA (TE) and the corresponding *P*-value, are shown.

Fig. 10a, b). Furthermore, we isolated poly(A) RNA from plant tissue samples of equal mass along with a synthesized poly(A) RNA spike-in control and quantified the ratio of poly(A) RNA to spike-in RNA by total RNA Pico Chip analysis using an Agilent 2100 Bioanalyzer. The *vir-1* mutant seedlings accumulated slightly higher levels of poly(A) RNA than wild-type seedlings, although this difference was not significant (Supplementary Fig. 10c), which verified that the m⁶A mediated by VIR does not significantly affect overall transcription.

An analysis of differences for the translatome (FDR < 0.05 and fold-change > 2) revealed that the number of transcripts with lower translation is markedly higher in the *vir-1* mutant. Specifically, we detected 268 genes with higher translation and 1762 genes with lower translation levels in *vir-1* before HL treatment (Supplementary Fig. 12c, j, k, and Supplementary Data 10). These numbers rose to 504 genes with higher translation and 1,990 genes with lower translation in *vir-1* upon HL exposure (Fig. 5d, e, Supplementary Fig. 12d, and Supplementary Data 11). GO enrichment analysis indicated that these differentially translated genes are enriched in a variety of basic growth and developmental processes before HL treatment (Supplementary Fig. 12l). However, the term response to high light intensity became greatly enriched among differentially translated genes after HL treatment (Fig. 5f). In addition, we noticed differences in transcript and translation levels between Col-0 and *vir-1* by principal component analysis (Supplementary Fig. 13a, b). The functions of these differentially expressed genes and differentially translated genes clearly aligned with the observed HL hypersensitivity of the *vir-1* mutant, suggesting that their underlying processes are modulated by VIR.

We therefore analyzed translation efficiency (TE) between Col-0 and *vir-1* before and after HL treatment by determining the ratio of translating mRNAs to total mRNAs. Compared to Col-0, we identified 96 upregulated differential TE genes and 359 downregulated differential TE genes (FDR < 0.05 and fold-change > 2) in *vir-1* before HL treatment (Supplementary Fig. 12e, m, n and Supplementary Data 12) and 110 upregulated differential TE genes and 385 downregulated differential TE genes in *vir-1* after HL treatment (Fig. 5g, h, Supplementary Fig. 12f, and Supplementary Data 13). The enriched GO terms for these differential TE genes included response to stimulus and response to stress (Fig. 5i and Supplementary Fig. 12o). Interestingly, in contrast to the strong correlation between mRNA abundance and translation observed within the same samples (Fig. 5j and Supplementary Fig. 13c), we observed a poor correlation for the changes between Col-0 and *vir-1* in mRNA abundance and translation (*r* = 0.15 before HL, *r* = 0.18 after HL) (Fig. 5k and Supplementary Fig. 13d), suggesting that VIR modulates gene expression via affecting both mRNA abundance and translation. For example, the effects of VIR on RNA-seq, Ribo-seq, and TE for the two key regulators of PSII repair, MPH1 and STN8, were distinct (Fig. 5l, m). There was a 40% decrease in *MPH1* mRNA abundance and a similar decrease in *MPH1* translation, but TE was similar in *vir-1* mutant and wild-type seedlings. By contrast,

*STN8* translation dropped by 45%, with a decrease in TE of 49%, although there was no change in *STN8* mRNA abundance.

## VIR-mediated m⁶A modification is required for stability and translation of photoprotection-related mRNAs

To further explore the influence of m⁶A modification mediated by VIR on photoprotection-related protein expression, we performed immunoblotting with anti-HHL1, anti-MPH1, anti-HCF244, anti-Deg1, anti-cpTatC, and anti-STN8 antisera. The levels of HHL1, MPH1, HCF244, and Deg1, and STN8 were all significantly lower in *vir-1* relative to Col-0 both before and after 24 h of HL treatment (Fig. 6a). Before HL treatment, the levels of HHL1, MPH1, HCF244, Deg1, and STN8 in *vir-1* were ~27%, 25%, 65%, 40%, and 43% of Col-0 levels, respectively; after HL treatment, they were ~30%, 18%, 65%, 43%, and 31% of Col-0 levels, respectively (Fig. 6b). The abundance of cpTatC in *vir-1* decreased only slightly after 24 h of HL treatment (Fig. 6a, b). The molecular defects seen in the *vir-1* mutant were rescued in the *VIR* complemented line (Supplementary Fig. 14).

We further explored how VIR affects photoprotection-related gene expression. m⁶A can affect the stability of mRNAs in mammals and plants[36,37,56]. We thus measured RNA levels for the two key PSII repair factor genes *HHL1*[48] and *MPH1*[49,57], whose accumulation was lower in the *vir-1* mutant (Fig. 6c), in Col-0 and *vir-1* seedlings incubated with the transcription inhibitor actinomycin D. The mRNAs of both *HHL1* and *MPH1* were more rapidly degraded in *vir-1* than in Col-0 seedlings (Fig. 6d), indicating that lower levels of the m⁶A modification promotes mRNA degradation. We obtained the same result for other photoprotection-related transcripts such as those for *CPRabA5e*, *Tha4*, and *FtsH2* (Supplementary Fig. 15). These molecular defects were rescued back to Col-0 levels in the *VIR* complemented line (Supplementary Fig. 15). The m⁶A modification can stabilize mRNAs by inhibiting local ribonucleolytic cleavage[44]. We explored whether the lower mRNA density and faster degradation in *vir-1* resulted from local ribonucleolytic cleavage. Based on the predicted motifs of the highest-cleaved nucleotide within m⁶A peaks[44], we compared the cleavage levels around motifs of the highest-cleaved nucleotide within different-level m⁶A peaks in *vir-1* and Col-0 plants by quantitative real-time PCR (qRT-PCR). The relative mRNA amounts of *HHL1* and *MPH1* containing the cleavage sites (RRACH motif) decreased by 22% and 17%, respectively, in *vir-1* compared to Col-0 (Fig. 6e, f), suggesting that mRNAs of *HHL1* and *MPH1* containing the RRACH motif in *vir-1* mutants are cleaved more than that in Col-0 plants. In addition, analysis of the translatome revealed that the total translation levels of *HHL1* are lower in *vir-1* relative to Col-0 (Fig. 6g) due to decreased *HHL1* mRNA abundance, although the TE did not change (Supplementary Fig. 16).

We analyzed the m⁶A sites in *HHL1* transcripts and identified three RRACH motifs around the stop codon (Fig. 6h). We thus constructed expression vectors harboring *HHL1* with point mutations in which the adenine residues within the m⁶A motifs are replaced by guanine (A-G transition mutations) (Fig. 6h). We generated four mutants with one

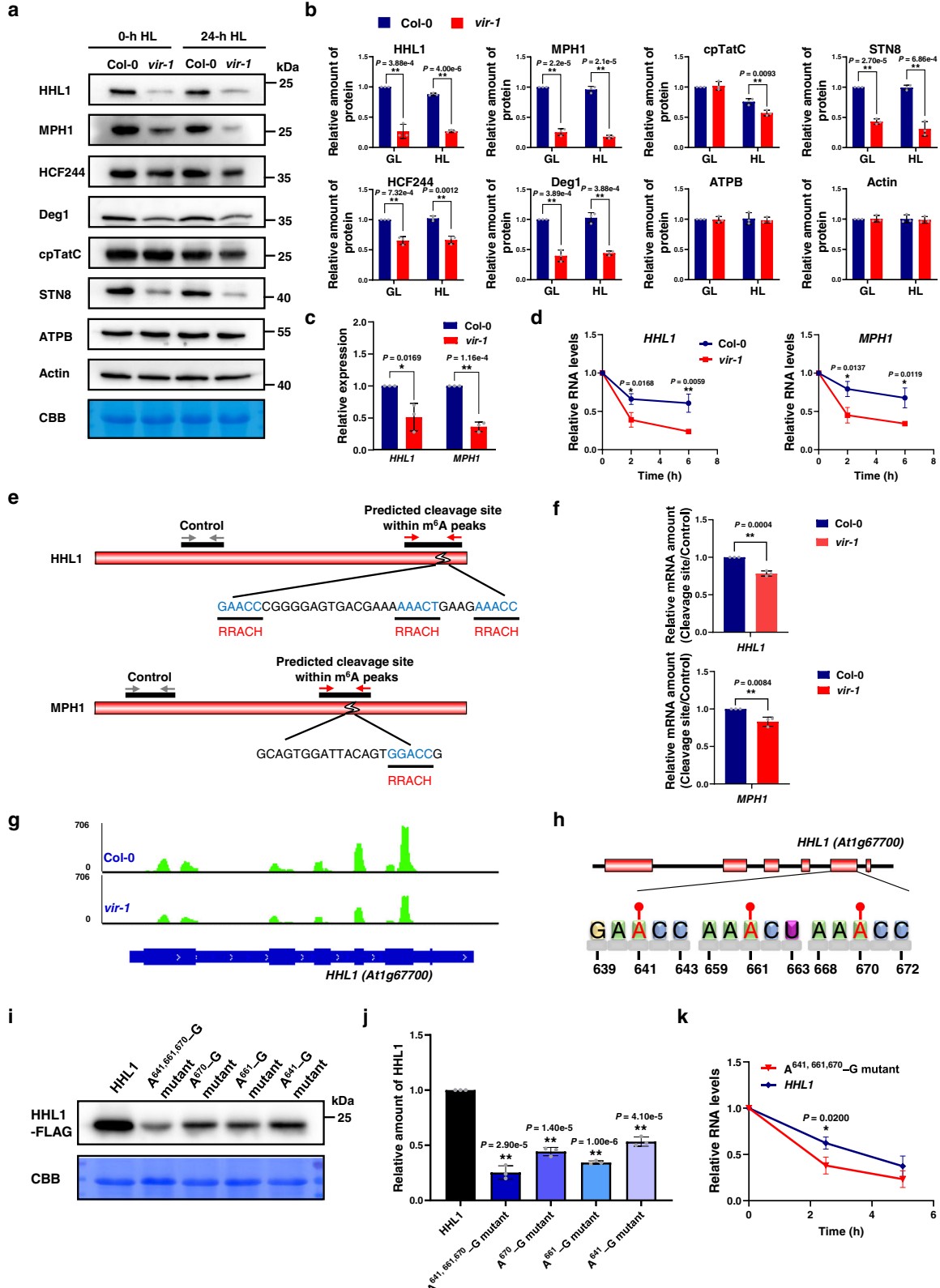

(A641-G, A661-G, and A670-G) or all three (A641,661,670-G) m⁶A motifs disrupted by replacing normally methylated adenine residues with guanine. We transformed *hhl1* protoplasts with equal amounts of plasmids overexpressing wild-type or mutant *HHL1*. Immunoblotting with anti-FLAG antisera indicated that HHL1 abundance decreases upon the removal of the m⁶A motif, to ~53%, 34%, 44%, and 25% of wild-type levels, in protoplasts expressing the A641-G, A661-G, A670-G, and

A641,661,670-G mutant constructs, respectively (Fig. 6i, j). We also analyzed the effects of the A641,661,670-G mutant on protein and mRNA stability. The A641,661,670-G mutant did not affect protein stability (Supplementary Fig. 17), but lowered mRNA stability (Fig. 6k). These results suggest that the alteration of both mRNA stability and translation by the A641,661,670-G mutations in the *HHL1* transcript contributes to the decreased HHL1 protein abundance.

**Fig. 6 | Effects of VIR-mediated m⁶A modification on the stability and translation of photoprotection-related gene mRNAs. a** Analysis of protein abundance in Col-0 and *vir-1* mutants. Protein samples from Col-0 and *vir-1* seedlings were separated by 12% SDS-urea-PAGE and probed with antisera against specific proteins. CBB was used to estimate loading. Similar results were obtained from three independent biological replicates. **b** Proteins immunodetected from (**a**) were quantified with Phoretix 1D Software (Phoretix International, UK). Values (means ± SE; $n = 3$ independent biological replicates) are given relative to protein levels of Col-0 before HL treatment. *$P < 0.05$; **$P < 0.01$, by two-sided Student's *t* test. **c**, Relative mRNA levels of *HHL1* and *MPH1* in Col-0 and *vir-1* seedlings. *UBQ10* was used as an internal control. Values are means ± SE ($n = 3$ biological replicates). *$P < 0.05$; **$P < 0.01$, by two-sided Student's *t* test. **d** mRNA lifetimes of *HHL1* and *MPH1* in Col-0 and *vir-1* seedlings. Seven-day-old Col-0 and *vir-1* seedlings treated with actinomycin D for 0, 2, or 6 h were used for transcription inhibition assays. 18S ribosomal RNA was used as the internal reference. Values are means ± SE ($n = 3$ biological replicates). *$P < 0.05$; **$P < 0.01$, by two-sided Student's *t* test. **e** Predicted cleavage site within m⁶A peaks in *HHL1* and *MPH1* transcripts and primer design. **f** Ratio of relative mRNA amounts at cleavage sites and control sites in Col-0 and *vir-1*. Values

are means ± SE ($n = 3$ biological replicates). *$P < 0.05$; **$P < 0.01$, by two-sided Student's *t* test. **g** Normalized distribution of Ribo-seq reads in Col-0 and *vir-1* along *HHL1*. **h** Schematic representation of the positions of m⁶A motifs within *HHL1*. **i** Analysis of HHL1-FLAG accumulation in *hhl1* protoplasts transfected with equal amounts of plasmids overexpressing wild-type or mutant *HHL1*. A⁶⁴¹,⁶⁶¹,⁶⁷⁰-G mutant, mutant harboring the transition mutations A641, A661, A670-G; A⁶⁷⁰-G mutant, mutant with A670-G transition mutation; A⁶⁶¹-G mutant, mutant with A661-G transition mutation; A⁶⁴¹-G mutant, mutant with A641-G transition mutation. **j** Proteins immunodetected from (**i**) were quantified with Phoretix 1D Software (Phoretix International, UK). Values (means ± SE; $n = 3$ independent biological replicates) are given relative to protein levels of wild-type HHL1. *$P < 0.05$; **$P < 0.01$, by two-sided Student's *t* test. **k**, Analysis of *HHL1* mRNA lifetimes in *hhl1* protoplasts transfected with equal amounts of wild-type or mutant *HHL1* plasmids. The overnight cultured protoplasts were treated with actinomycin D for 0, 2.5, or 5 h for transcription inhibition assays. 18S ribosomal RNA was used as the internal reference. Values are means ± SE ($n = 3$ biological replicates). *$P < 0.05$; **$P < 0.01$, by two-sided Student's *t* test.

---

We transformed *vir-1* protoplasts with plasmids overexpressing wild-type or mutant *HHL1* to assess whether the effects of *HHL1* m⁶A sites on protein abundance are VIR-dependent. HHL1 abundance only decreased to ~67% of wild-type levels in *vir-1* protoplasts expressing the A641,661,670-G mutant construct (Supplementary Fig. 18), but decreased to ~25% of wild-type levels in *hhl1* protoplasts expressing A641,661,670-G mutants (Fig. 6j). This result suggests that the effect of *HHL1* m⁶A sites on protein abundance is VIR-dependent.

## VIR associates with photoprotection-related mRNAs and proteins involved in post-transcriptional regulation

To understand the molecular mechanism of VIR in regulating photoprotection-related mRNAs, we performed RNA immunoprecipitation (RIP) assays and pulled down VIR-associated mRNAs and proteins (Fig. 7a). First, we determined that VIR associates with the photoprotection-related *HHL1*, *MPH1*, *HCF244*, *Deg1*, *cpTatC*, and *STN8* transcripts in vivo (Fig. 7b, c). *ACTIN*, used as a negative control, was barely detected. This result suggests that VIR regulates the mRNA stability or translation of these genes by direct binding.

Furthermore, we identified 182 VIR-associated proteins by LC-MS/MS analysis of the RIP assays (Fig. 7d and Supplementary Fig. 19b). GO enrichment analysis revealed that these proteins are mainly enriched in molecular functions related to 'structural constituent of ribosome', 'RNA binding', and 'protein binding', etc. (Fig. 7e); and biological processes related to 'translation', 'photosynthesis', 'response to stress', and 'embryo development', etc. (Supplementary Fig. 19c). Intriguingly, this analysis also identified important VIR interactors, including the m⁶A writer HAKAI, and regulators of RNA stability, processing and translation, such as the Arabidopsis Eukaryotic translation initiation factor 3 (eIF3e), ribonuclease J (RNJ), DEAD box RNA helicase 3 (RH3), and DEAD box RNA helicase 6 (RH6) (Fig. 7f). Notably, eIF3e participates in translation[58]; RNJ functions in the regulation of RNA processing and RNA stability[59,60]; and RH3 and RH6 are required for RNA processing and translation[61,62]. Other factors involved in chloroplast gene expression, such as plastid transcriptionally active 4 (PTAC4), PTAC5, PTAC16 and the RNA Recognition Motif (RRM)-containing proteins UBA2C and ATG3BP6 were also co-immunoprecipitated (Fig. 7f). Our results suggest that VIR associates with photoprotection-related mRNAs and proteins involved in post-transcriptional regulation.

## Discussion

m⁶A, the most common post-transcriptional modification of transcripts in eukaryotes, participates extensively in post-transcriptional regulation[23]. The m⁶A modification of transcripts has been reported to function in plant growth, development[31,33,34,36], and salt stress

response[44,63,64]. The m⁶A modification regulates the mRNA stability of transcripts in response to salt stress[44,63,64]. However, the roles of m⁶A in photosynthesis and light stress response are unknown. The contribution of m⁶A-mediated translational regulation in response to stress remains unclear in plants, although m⁵C modification was reported to be required for plant adaptation to heat stress by regulating mRNA translation[65]. In this study, we systematically analyzed the role of m⁶A-mediated post-transcriptional regulation in the maintenance of photosynthesis under high light stress conditions. We demonstrated that m⁶A-regulated mRNA stability and translation are required for maintaining photosynthetic efficiency under high light (HL) stress conditions in plants (Fig. 7g).

HL can result in photodamage of photosystems (especially PSII), thus reducing photosynthetic efficiency[2]. To prevent photodamage and to acclimate to changes in their environment, plants have evolved mechanisms for sensing and responding to HL[12], such as modulating the expression of photoprotection-related genes[20]. Dot blot and LC-MS analyses showed that HL induced the overall m⁶A level in wild-type Arabidopsis plants (Fig. 1a, b), which is consistent with the upregulated expression levels of genes encoding m⁶A writers after HL treatment (Fig. 1g), suggesting that HL induces an increase of overall m⁶A levels in mRNA. We further checked m⁶A modification levels using m⁶A-seq, according to a published method[66] that is often followed[24,33,67]. We detected numerous m⁶A modification sites in chloroplast/photosynthesis-related genes (Supplementary Fig. 9), the levels of some of which changed upon exposure to HL treatment (Fig. 1d). Chloroplast/photosynthetic transcripts also contain the m⁶A modification, although the effect of m⁶A on chloroplast function and photosynthesis, and the underlying regulatory mechanism remain unknown[24]. HL induced changes in m⁶A modification levels (Fig. 1e, f), suggesting that m⁶A modification dynamically responds to high light stress. Surprisingly, m⁶A-seq showed that more m⁶A sites were downregulated than upregulated, suggesting that the overall increased abundance of the m⁶A modification exceeded the overall decreased abundance of the m⁶A modification.

In a recent study, differential error sites from nanopore direct RNA sequencing (DRS) were used to identify transcriptome-wide m⁶A modifications at single-base resolution[68]. Both this and the previous study observed that many m⁶A peaks contained the m⁶A consensus motif RRACH (R = A/G, H = A/C/U), mainly at the 3′ end of transcripts. Due to the different detection accuracy of the two techniques, m⁶A sites were mainly enriched in 3′ UTRs in the previous study[68], while we preferentially detected m⁶A sites near the stop codon in our study, which was consistent with other published results[24,33,36]. We determined that *HHL1*, an important photoprotective gene, harbors an obvious m⁶A peak near its stop codon (Fig. 4e), in a region that

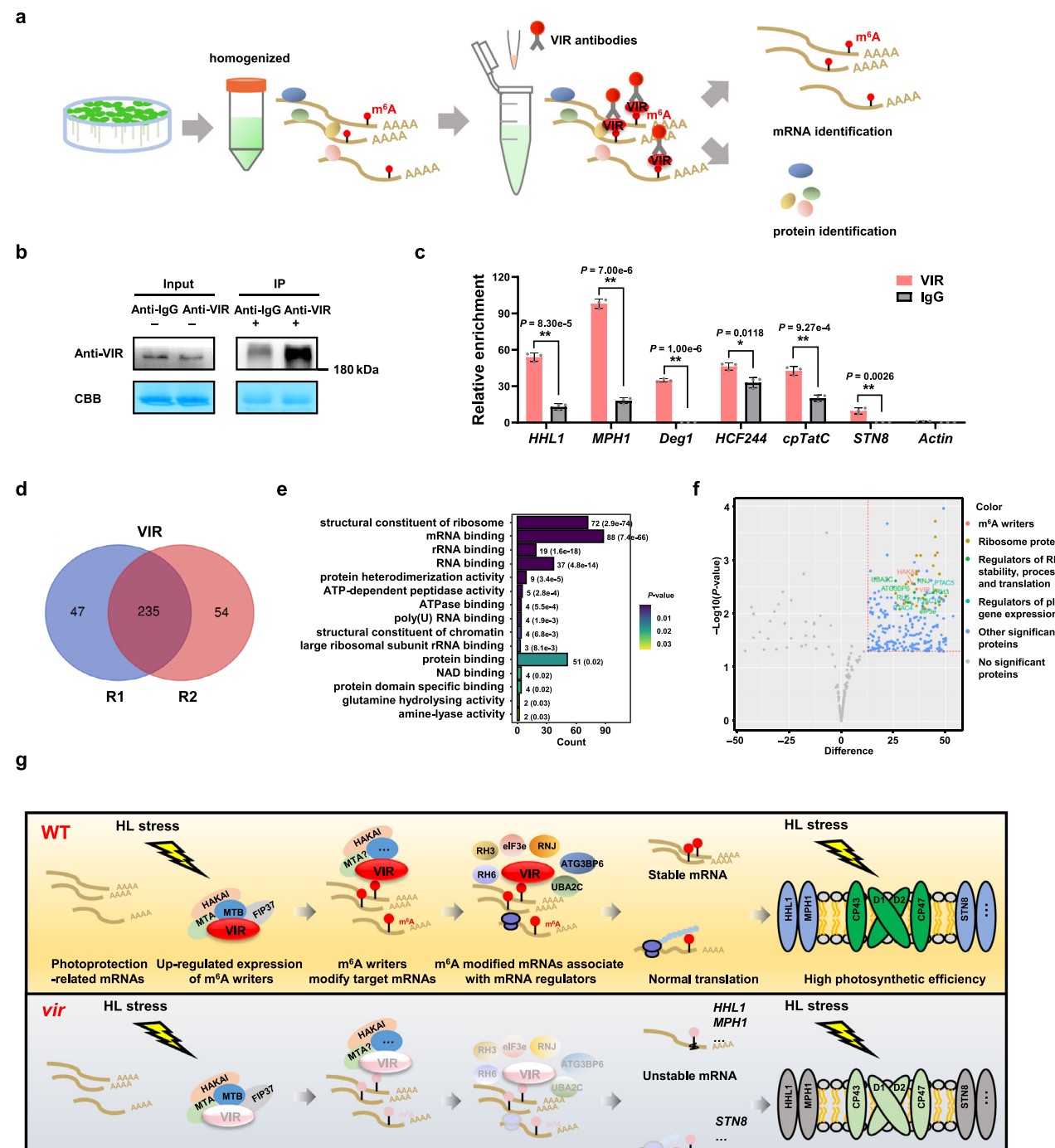

includes three RRACH motifs (Fig. 6h). When we replaced the adenines in the m6A motifs with guanine, the accumulation of HHL1 protein decreased by 47–75% (Fig. 6i, j), suggesting that the m6A modifications near the stop codon of *HHL1* are important for HHL1 translation.

Several lines of evidence support a vital positive role for the m6A writer VIR in maintenance of photosynthetic efficiency under high light stress. First, analysis of chlorophyll fluorescence showed that the *vir-1* mutant has a much lower photosynthetic activity and electron transport rate compared to the wild type after HL treatment (Fig. 2 and Supplementary Table 1), suggesting that VIR is required for maintenance of photosynthetic efficiency under HL conditions. Second, BN-PAGE and immunoblot analyses highlighted the dramatically reduced levels of PSII complexes and subunits after HL treatment (Fig. 3), which is consistent with PSII being the main site of photodamage[13]. These

results support the notion that VIR is involved in PSII photoprotection. In addition, the partial loss of VIR function also led to a decreased abundance for other complexes, including a slight decrease in PSI and a significant decrease in cyt *f* levels, which may also in part contribute to the high light hypersensitivity of the *vir-1* mutant. A recent study indicated that cyt *f* is also a target of photodamage and plays an important role in photoprotection[69], suggesting a general role for VIR in the photoprotection of photosystems. Notably, *VIR* was strongly expressed during the early stages of growth and in the aerial parts of plants (Supplementary Fig. 4b), implying that it plays an important role in young photosynthetic tissue, which is similar to the proposed role for CHLOROPHYLLASE1 in protecting young leaves from long-term photodamage[70]. In addition, the $F_V/F_m$ values of *ABI3:MTA* were lower than those in the control (Supplementary Fig. 3), implying that *vir-1*

**Fig. 7 | The interaction of VIR with photoprotection-related mRNAs and mRNA regulators. a** Schematic illustration of the experimental design. RNA immuno-precipitation (RIP) assays coupled to qPCR analysis and liquid chromatography-tandem mass spectrometry (LC-MS/MS) analysis identified mRNA and proteins pulled down by VIR. **b** Immunoblot analysis using anti-VIR antibody showing the accumulation of VIR in the input fraction and the immunoprecipitated fraction (Anti-VIR-IP) from Col-0 seedlings. **c**, RIP-qPCR assays showing that VIR directly binds to *HHL1*, *MPH1*, *HCF244*, *Deg1*, *cpTatC*, and *STN8* transcripts in vivo. *TUB2* was used as an internal control. Values are means ± SE (*n* = 3 biological replicates). *$P$ < 0.05; **$P$ < 0.01, by two-sided Student's *t* test. **d** Overlap in proteins identified in biological replicates (R1-R2) of VIR samples. **e** GO enrichment analysis of the molecular functions of 182 VIR-associated proteins. Statistical test was determined by one-sided hypergeometric test. **f** Volcano plot of most significant proteins identified by RIP pull-down assays. Red line (significance, 0.05) separates VIR-associated proteins (top right portion of plot) from background. The plots were constructed by the *t* test difference versus the negative log of the *P*-values for each protein. Statistical test was determined by two-sided Student's *t* test. Different

Selec*t*ed top hits are indicated with red dots for m⁶A writer, dark yellow dots for ribosome proteins, green dots for regulators of RNA stability, processing, and translation, blue-green dots for regulators of plastid gene expression, and blue dots for other significant VIR-associated proteins. Specific interactors are listed in Supplementary Table 5. **g** Schematic model of VIR-dependent m⁶A participation in photoprotection in plants. In wild-type (WT) Arabidopsis plants, high light (HL) stress induce upregulated expression of m⁶A writers, including VIR, MTA, MTB, HAKAI, and FIP37. VIR can associate with photoprotection-related mRNAs and modify their m⁶A modifications, which recruit mRNA regulators, which promotes their stability (such as *HHL1* and *MPH1*), and translation efficiency (such as *STN8*) and thus improves photoprotection capacity and photosynthetic efficiency under excess light conditions. In *vir-1* mutants, VIR deficiency results in defects of m⁶A modifications of photoprotection-related mRNA, which fail to associate with mRNA regulators, and decreases their stability due to ribonucleolytic cleavage (such as *HHL1* and *MPH1*) and translation efficiency (such as *STN8*), and reduces photo-protection capacity and photosynthetic efficiency under excess light.

---

may affect a portion of mRNA, while other writers also contribute to photoprotection in plants. Meanwhile, another m⁶A writher, HAKAI[34], interacted with VIR (Fig. 7f), supporting the idea that the m⁶A regulators may interact in m⁶A modification of photoprotection-related transcripts.

The m⁶A-seq analysis revealed that the different m⁶A peaks between Col-0 and *vir-1* were significantly enriched in genes associated with the chloroplast. It also involved the chloroplast epitranscriptome of the *vir-1* mutant in a previous study, which mainly focused on the technicalities of using Nanopore sequencing to detect m⁶A modifications at a single-base resolution[68]. The partial loss of VIR function affected the m⁶A modification of photoprotection-related transcripts. The transcripts of numerous photoprotection-related genes, including *HHL1*, *MPH1*, *HCF244*, *Deg1*, *cpTatC*, and *STN8*, showed strongly reduced levels of m⁶A modification in the *vir-1* mutant compared to Col-0 under both normal and HL conditions (Supplementary Fig. 8g and Fig. 4e). RIP analysis further revealed that VIR directly associated with the mRNAs of *HHL1*, *MPH1*, *HCF244*, *Deg1*, *cpTatC*, and *STN8* (Fig. 7b, c). These results suggest that VIR mediates the m⁶A methylation of transcripts derived from these photoprotective genes. Immunoblot analysis demonstrated a drastic reduction in HHL1, MPH1, HCF244, Deg1, and STN8 levels in the *vir-1* mutant (Fig. 6a, b), suggesting that VIR is required for the accumulation of these photo-protective proteins. Notably, mutants in *HHL1* and *MPH1* are hypersensitive to high light[48,49], as is the *vir-1* mutant. Under normal conditions, the photochemical efficiency was similar between the *vir-1* mutant and Col-0, although the m⁶A and protein levels of photoprotection-related transcripts significantly decreased without HL treatment, which was similar to that in the mutants of photoprotection-related genes, such as *HHL1*[48] or *MPH1*[49], because photoprotection factors mainly function in photodamage from HL stress. The transcriptome and translatome showed that VIR regulates the levels of HHL1 and MPH1 proteins by mainly controlling the abundance of their respective mRNAs (Supplementary Fig. 16). The mRNA stability assay established that the lower mRNA accumulation levels of *HHL1* and *MPH1* in *vir-1* can be attributed to the faster degradation of these mRNAs in *vir-1* relative to Col-0 seedlings (Fig. 6c, d), suggesting that the m⁶A modification mediated by VIR is involved in the maintenance of mRNA stability of photoprotective genes. The m⁶A modification inhibits local ribonucleolytic cleavage and thus stabilizes mRNAs in plants, which is required for salt and osmotic stress responses[44]. VIR interacts with RNJ, which functions in the regulation of RNA processing and RNA stability and is required for chloroplast and embryo development[59,60,71]. Moreover, more *HHL1* and *MPH1* mRNAs containing the RRACH motif were cleaved in *vir-1* mutants than in Col-0 plants (Fig. 6f), demonstrating the general mechanism of m⁶A in stabilizing mRNAs in the stress response.

Decreased mRNA abundances due to faster degradation ultimately resulted in reduced translation (such as that of *HHL1* and *MPH1*). In addition, VIR also specifically regulated the translation efficiency of photoprotection-related genes, but did not affect their mRNA stability (such as that of *STN8*; Fig. 5m and Supplementary Fig. 16). Interestingly, translation-related proteins, especially ribosome protein, were most enriched among VIR-associated proteins (Fig. 7f). eIF3e, a translation initiation factor, which promotes binding of the 43 S preinitiation complex to the 5′ end of mRNA[72], interacted with VIR. VIR also interacted with the RNA helicases RH3 and RH6, which regulate RNA processing and translation[61,62]. In addition, the RRM-containing proteins UBA2C and ATG3BP6 were also co-immunoprecipitated, suggesting the general role of VIR in post-transcriptional regulation.

Transcriptome-wide m⁶A profiling showed that, in addition to the five photoprotection-related genes mentioned above, many other photoprotection-related genes have greatly diminished levels of m⁶A modification in the *vir-1* mutant. These additional targets of VIR included the genes encoding filamentation temperature-sensitive protein Y (cpFtsY)[73], chloroplast Rab GTPase A5e (CPRabA5e)[74], chloroplast secretory translocase E (cpSecE)[52], CYO1[75], High Chlor-ophyll Fluorescence Phenotype 173 (HCF173)[76], Low Photosystem II Accumulation 1 and 2 (LPA1 and LPA2)[77], 20-kDa cyclophilin 3 (CYP20-3)[78], 20-kDa FK506-binding protein (FKBP20-2)[79], state transi-tion 7 (STN7)[53], Degradation of periplasmic proteins 2, 5, and 7 (Deg2, Deg5, and Deg7)[80], filamentation temperature-sensitive proteins H 1 and 2 (FtsH1 and FtsH2)[81], and photosystem II proteins 28 and 29 (PSB28 and PSB29)[82] (Supplementary Fig. 9a, b), which function in the biogenesis, assembly, stability, and repair of the photosystems[83]. These results suggest that the high light hypersensitivity of the *vir-1* mutant may be due to the combined effects of numerous target genes, and VIR may constitute a hub that mediates a complex photoprotec-tion network in multiple post-transcriptional processes, which is similar to the role of LONG HYPOCOTYL5 (HY5) in the transcriptional regulation of photoprotection-related genes[21]. It is possible that more targets of VIR involved in photoprotection can be detected by nano-pore direct RNA sequencing due to the higher detection accuracy in a recent report[68]. Interestingly, the levels of VIR-mediated m⁶A mod-ification in some transcripts of photoprotection-related genes did not change significantly between normal and HL conditions (Supplementary Fig. 9a, b, Fig. 4e, and Supplementary Fig. 8g), suggesting that the m⁶A modification of these photoprotection-related transcripts is written by VIR in a constitutive manner.

Transcriptional regulation is an important response to HL stress, and the transcription of photoprotection-related genes is clearly induced under HL[21,84,85]. Interestingly, we discovered that the expres-sion of *VIR* and other m⁶A regulator genes was strongly induced by HL (Fig. 1g, h), although the VIR-mediated m⁶A modification did not

significantly affect overall transcription (Supplementary Fig. 10). Light is a vital environmental signal that is perceived by photoreceptors[86] that modulate the regulation of gene expression mediated by transcription factors, thereby affecting plant growth, development, and stress responses[21,87]. The apparent upregulation of m6A regulator genes (especially m6A writers), under HL conditions may be caused by the activation of photoreceptor-mediated light signaling cascades. It is also possible that reader proteins may read these m6A sites, or other m6A-related mechanisms may affect transcript levels, and future pull-down experiments with the m6A-modified RNA probes could help identify reader proteins that bind these target transcripts. Thus, our study uncovered a possible link between transcriptional regulation mediated by light signaling and post-transcriptional regulation mediated by m6A modification in response to photodamage stress in plants.

## Methods

### Plant materials and growth conditions

All Arabidopsis (*Arabidopsis thaliana*) lines used in this study were in the Columbia-0 (Col-0) background. Seeds of the *vir-1* mutants (*VIR*: At3g05680), *VIRprom:GFP-VIR* complementation lines, and *UBQ10prom:XVE»MTB* RNAi (*MTB*-RNAi) RNAi knockdown lines have been previously described[34]. The *ect2-1*, *ect3-1*, and *ect4-2* mutants[38] were obtained from the Nottingham Arabidopsis Stock Center (NASC) (SALK_002225, SALK_077502, and GK-241H02). The *mta ABI3-prom:MTA* complemented line (*mta* SALK_074069) (*ABI3:MTA*)[32] was kindly provided by Prof Rupert Fray. The β-estradiol-inducible transgene *UBQ10prom:XVE»VIR* RNAi (*VIR*-RNAi-1 and *VIR*-RNAi-2) constructs were generated as described previously[34], inserting the regions detailed in Supplementary Table 4. Arabidopsis seedlings were grown on soil in a growth chamber (120 µmol photons m$^{-2}$ s$^{-1}$, 12-h light/12-h dark photoperiod, 21 °C/18 °C day/night, and 60% relative humidity). *MTB*-RNAi lines were grown on half-strength Murashige and Skoog (MS) medium containing 5 µM β-estradiol (Sigma). For the *VIR* RNAi line (*VIR*-RNAi-1 and *VIR*-RNAi-2) and the corresponding control, plant materials were first grown without β-estradiol for 4 days, then sprayed with 50 µM β-estradiol for 3 days. For high light treatment, 7-day-old seedlings were transferred to a growth chamber and grown under continuous high light (1000 µmol photons m$^{-2}$ s$^{-1}$, 17 °C, and 60% relative humidity) for 0, 4,12, and 24 h.

### Dot blot analysis of m6A levels

Poly(A) RNA was purified from total RNA using a Dynabeads™ mRNA purification kit (Thermo Fisher, USA) and then the purified mRNA was heated at 95 °C for 3 min. The mRNA was then used for Dot blot[33]. The mRNA was spotted on an Amersham Hybond-N + membrane. After UV crosslinking, the membrane was blocked with 5% (w/v) non-fat milk in Tris-buffered saline with Tween-20 (TBST), and incubated with anti-m6A antibody (1:500; Synaptic Systems) overnight at 4 °C. After incubation with a secondary anti-rabbit antibody, the membrane was visualized using SuperSignal West Pico chemiluminescent substrate (Thermo Scientific).

### Quantification of m6A levels in mRNA by LC-MS/MS

First, 200 ng Col-0 mRNA was digested into nucleosides by Nuclease P1 (NEB and shrimp alkaline phosphatase (NEB) in 50 µl RNase-free water, and incubated at 37 °C overnight. The mixture was diluted to 100 µl, and 10 µl from each sample was injected into an LC-MS/MS system consisting of a high-performance liquid chromatographer (Shimadzu) equipped with a C18-T column (Weltech) and a Triple Quad 4500 (AB SCIEX) mass spectrometer in positive ion mode by multiple-reaction monitoring. Mass transitions of $m/z$ 268.0–136.0 (A), $m/z$ 245.0–113.1 (U), $m/z$ 244.0–112.1 (C), $m/z$ 284.0–152.0 (G), and $m/z$ 282.0–150.1 (m6A) were monitored and recorded. The concentration of nucleosides was quantified according to the standard curves generated against pure commercial nucleosides (MCE).

### Chlorophyll fluorescence

Chlorophyll fluorescence parameters were measured with the MAXI version of the IMAGING-PAM M-Series chlorophyll fluorescence system (Heinz-Walz Instruments)[88]. To measure light-response curves of PSII quantum yield (ΦPSII), electron transport rate (ETR), and non-regulated non-photochemical quenching yield [Y(NO)], seedlings were illuminated at the following light intensities: 0, 24, 83, 130, 192, 264, 348, 444, 671, and 908 µmol photons m$^{-2}$ s$^{-1}$. The duration of illumination for each light intensity was 3 min. At the end of the 3-min period, a saturation pulse was applied and the value was recorded. For Non-photochemical quenching (NPQ) measurements, actinic light (500 µmol photons m$^{-2}$ s$^{-1}$) was turned on at time zero after initial determination of $F_0$ and $F_m$, and the seedlings were kept in the dark for 8 min after 12 min of treatment. At the end of the 1-min period, a saturation pulse was applied and the value was recorded[48].

### Blue native (BN)/SDS-PAGE and immunoblot analysis

BN-PAGE was performed as described[77]. Thylakoid membranes were quantified based on chlorophyll contents. The thylakoid membrane preparations were solubilized in 2% (w/v) n-dodecyl-β-D-maltoside (DM) and incubated at 4 °C for 20 min, centrifuged at 12,000 g for 10 min at 4 °C, and loaded onto a 5% to 13.5% gradient acrylamide gel. For SDS-PAGE, protein samples were separated on 12% (w/v) SDS-urea-PAGE gels. After electrophoresis, the proteins were transferred to polyvinylidene difluoride membranes (Millipore) and probed with antibodies. The following antisera against photosynthetic proteins were purchased from Agrisera and PhytoAB: D1, AS05084; D2, AS06146; CP43, AS111787; CP47, AS04038; PsbO, AS05092; PsaB, AS10695; Cyt $f$, AS08306; LHCA1, AS01005; LHCB1, AS01004; ATPB, AS05085; MPH1, PHY1234A; HCF244, PHY0327; Deg1, PHY0146A; cpTatC, PHY2198S. The HHL1 antibody was from[48]. SuperSignal West Pico chemiluminescent substrate (Thermo Scientific) was used to generate signals, and the signals were detected with a chemiluminescence system.

### Analysis of protein degradation

Analysis of protein degradation was performed as described[48]. In brief, several detached leaves were placed with their adaxial side up on filter paper soaked with sodium phosphate buffer (pH 7.0) and illuminated at 1,000 µmol photons m$^{-2}$ s$^{-1}$. Other detached leaves were incubated in buffer containing 200 µg mL$^{-1}$ chloramphenicol under reduced pressure for 30 min before being exposed to photoinhibitory light treatment. To analyze the accumulation of PSII proteins under high light intensity, proteins extracted from leaves were subjected to SDS-PAGE and immunoblotting with antibodies against PSII proteins.

### Quantitative real-time polymerase chain reaction analysis

Total RNA was extracted from Arabidopsis seedlings using a Plant RNA Kit (Magen) according to the manufacturer's instructions. Total RNA (2 µg) was reverse-transcribed into first-strand cDNA using a Prime-Script RT Reagent kit (Takara) according to the manufacturer's instructions. Quantitative real-time PCR (qRT-PCR) was carried out using SYBR Premix ExTaq reagent (Takara), and real-time amplification was monitored on a LightCycler480 system (Roche). *ACTIN2* and *UBQ10* were used as the internal references. Primer sequences are listed in Supplementary Table 4.

### m6A sequencing and validation

Seven-day-old Col-0 and *vir-1* seedlings were collected immediately before (0 time point) and after 4 h of high light treatment. Total RNA was isolated from the seedlings and purified using TRIzol reagent (Invitrogen, USA) following the manufacturer's procedure. The level and purity of each sample was tested using a NanoDrop and RNase-free agarose gel electrophoresis. Then, m6A sequencing was performed[66]. Poly(A) RNA was purified from 2 mg total RNA using a Dynabeads™ mRNA purification kit (Thermo Fisher, USA) with two rounds of

purification. The purified poly(A) RNA was randomly fragmented into small fragments using RNA Fragmentation Reagents (Thermo Fisher Scientific, USA). The fragmented RNA was incubated for 2 h at 4 °C with m⁶A-specific antibody (No. 202003, Synaptic Systems, Germany) in IP buffer (50 mM Tris-HCl, pH 7.5, 750 mM NaCl, and 0.5% [v/v] IGEPAL CA-630). The mixture was then incubated with protein A beads (Thermo Fisher scientific, USA) that had been pre-blocked with bovine serum albumin (BSA) for an additional 4 h at 4 °C. After washing three times with IP buffer, bound RNA was eluted from the beads with elution buffer (6.7 mM N6-methyladenosine [Sigma-Aldrich] in IP buffer), followed by ethanol precipitation. A NEBNext® Ultra™ RNA Library Prep Kit for Illumina (NEB) was used to construct the library from immunoprecipitated RNA and input RNA. Sequencing was carried out on an Illumina NovaSeq platform.

To validate the m⁶A sequencing results, 7-day-old Col-0 and *vir-1* seedlings that had been subjected to 0 or 4 h of high light treatment were used for m⁶A immunoprecipitation. Poly(A) RNA was purified from 1 mg total RNA using a Dynabeads™ mRNA purification kit (Thermo Fisher, USA) with two rounds of purification. The poly(A) RNA was randomly fragmented into ~200-nt fragments using RNA fragmentation reagents (Thermo Fisher Scientific, USA). The fragmented RNA was incubated for 2 h at 4 °C with m⁶A-specific antibody (No. 202003, Synaptic Systems, Germany) in IP buffer (50 mM Tris-HCl, pH 7.5, 750 mM NaCl, and 0.5% [v/v] IGEPAL CA-630). The mixture was then incubated with protein A beads (Thermo Fisher Scientific, USA) that had been pre-blocked with BSA and eluted with elution buffer (6.7 mM N6-methyladenosine [Sigma-Aldrich] in IP buffer). The input RNA and immunoprecipitated RNA were reverse-transcribed using M-MLV Reverse Transcriptase (Thermo Fisher Scientific, USA). The m⁶A levels of specific mRNA fragments were determined by qRT-PCR, first normalized to *TUB2* levels, and then the ratio of the abundance of the IP sample against the input sample was calculated.

## RNA-seq

Seven-day-old Col-0 and *vir-1* seedlings were collected after 0 or 4 h of high light treatment. Total RNA was isolated from the seedlings and purified using TRIzol reagent (Invitrogen, USA) following the manufacturer's procedure. RNA quality was assessed on an Agilent 2100 Bioanalyzer (Agilent Technologies, Palo Alto, CA, USA) and checked by RNase-free agarose gel electrophoresis. Libraries for RNA-seq were prepared using a NEBNext® Ultra™ RNA Library Prep Kit for Illumina (NEB, USA) following the manufacturer's instructions. Sequencing was performed on an Illumina HiSeq 2500 platform by Gene Denovo Biotechnology Co. (Guangzhou, China).

## Quantitative analysis of poly(A) RNA

Poly(A) RNA spike-in control was transcribed in vitro using the MEGAscript kit (Thermo Fisher, USA)[55]. Total RNA was extracted from equal masses of Seven-day-old Col-0 and *vir-1* seedling after 4-h HL treatment using the TRIzol reagent (Invitrogen, USA) and was added with the spike-in control. Poly(A) RNA, along with the spike-in, was purified using a Dynabeads™ mRNA purification kit (Thermo Fisher, USA). The ratio of poly(A) RNA to spike-in RNA was quantified by total RNA Pico Chip analysis using an Agilent 2100 Bioanalyzer[55]. For quantitative RNA-seq, equal masses of Seven-day-old Col-0 and *vir-1* seedling after 4-h HL treatment were used. After total RNA isolation, External RNA Controls Consortium (ERCC) RNA spike-in control (Ambion) was added to each isolated total RNA sample[55]. A NEBNext® Ultra™ RNA Library Prep Kit for Illumina (NEB) was used to construct the library, Sequencing was performed by LC-Bio Technology CO., Ltd (Hangzhou, China).

## Ribosome profiling (Ribo-seq)

The 7-day-old Col-0 and *vir-1* seedlings were subjected to 0 or 4 h of high light treatment, immediately frozen in liquid nitrogen for at least 1 h, ground into a fine powder in liquid nitrogen, and dissolved in 400 μl of lysis buffer (20 mM Tris-HCl, pH 7.4, 150 mM NaCl, 5 mM MgCl₂, 1 mM DTT, 100 μg/ml cycloheximide, and 1% [v/v] Triton X-100). Following incubation on ice for 10 min, the lysate was centrifuged at 17,000 g for 10 min at 4 °C. To prepare ribosome footprints (RFs), 10 μl of RNase I and 6 μl of DNase I were added to 400 μl clarified lysate. After incubation for 45 min at room temperature, 10 μl of SUPERase-In RNase inhibitor was added to stop the nuclease digestion. The digested lysate was placed into a pre-equilibrated size exclusion column (Illustra MicroSpin S-400 HR Columns; GE Healthcare) and eluted from the column by centrifugation at 600 g for 2 min. An RNA Clean and Concentrator-25 kit (Zymo Research) was used to isolate RFs larger than 17 nt. A probe was used to remove rRNA, and the RFs were further purified with magnetic beads (Vazyme)[89]. NEBNext Multiple Small RNA Library Prep Set for Illumina (NEB) was used to construct the Ribo-seq libraries. Sequencing was performed on an Illumina HiSeq 2500 platform by Gene Denovo Biotechnology Co. (Guangzhou, China).

## Sequencing data analysis

m⁶A-seq data: Fastp[90] was used to remove the reads containing adaptor contamination, low-quality bases, and undetermined bases using default parameters. The sequence quality was also verified using fastp. The sequencing reads were mapped to the TAIR10 Arabidopsis reference genome using HISAT2 software[91]. Mapped reads of IP and input were fed into the R package exomePeak[92] to identify m⁶A peaks using default parameters and a cutoff *P*-value of 0.05. The differentially methylated m⁶A peaks (diffPeaks) between two sets of samples had to fulfill the criteria of false discovery rate (FDR) < 0.05 and enrichment fold-change ≥ 2. The consensus motif was determined using HOMER. Genes with differentially methylated m⁶A peaks were subjected to Gene Ontology (GO) enrichment and Kyoto Encyclopedia for Genes and Genomes (KEGG) pathway analyses.

RNA-seq data: Low-quality reads were removed and the adapter sequences were clipped with fastp[90]. Reads that mapped to rRNAs were removed, and the remaining unmapped reads were used for subsequent transcriptome analysis. The clean reads were mapped to the TAIR10 Arabidopsis reference genome using HISAT2.2.4[91]. Gene expression levels were estimated using StringTie (version 1.3.1)[93,94] and normalized using FPKM (fragments per kilobase of transcript per million mapping reads). Differential expression analysis was performed using DESeq2[95] software. Genes fulfilling the criteria FDR < 0.05 and fold-change > 2 were considered to be differentially expressed and were subjected to GO enrichment analysis.

Ribo-seq data: Raw reads containing >50% low-quality bases or >10% N bases were removed, and the adapter sequences were trimmed by Fastp[90]. Reads 10–50 bp in length were retained for subsequent analysis. The reads that mapped to ribosome RNAs (rRNAs), transfer RNAs (tRNAs), small nucleolar RNAs (snoRNAs), small nuclear RNAs (snRNAs), and microRNA (miRNAs) were removed. The remaining sequence reads were mapped to the TAIR10 Arabidopsis reference genome with Bowtie2[96], allowing no mismatches. Read numbers in the open reading frames of coding genes were calculated with RiboTaper software[97]. The FPKM method was used to normalize gene expression levels. The edgeR package (http://www.rproject.org/) was used to identify differentially translated genes across sample groups. Genes fulfilling the criteria of FDR < 0.05 and fold-change > 2 were considered to be differentially translated and were subjected to GO enrichment analysis.

## Translational efficiency (TE) analysis

Translational efficiency is the ratio of translating mRNAs (as FPKM from Ribo-seq) to total mRNAs (FPKM from RNA-seq) for a gene. The TE values were calculated and compared between samples and groups. RiboDiff[98] was used to identify differential TE genes across sample groups. Genes fulfilling the criteria FDR < 0.05 and fold-change

> 2 were considered to be differential TE genes and were subjected to GO analysis.

## Production of polyclonal anti-VIR antibodies

Affinity-purified anti-VIR polyclonal antibodies were generated by GenScript. A 15–amino acid peptide (corresponding to amino acids 1,552 to 1,565 of VIR) with an additional C-terminal Cys residue (APTRRDAFRQRKPNC) was synthesized, conjugated with keyhole limpet hemocyanin, and used to immunize rabbits and raise antibodies against VIR.

## RNA immunoprecipitation

The 7-day-old Col-0 seedlings were collected, and 2 g of tissue was ground to a fine powder in liquid nitrogen and homogenized in 2 ml of lysis buffer (50 mM Tris-HCl, pH 7.4, 2.5 mM $MgCl_2$, 100 mM KCl, 0.1% [v/v] Nonidet P-40, 1 μg/ml leupeptin, 1 μg/ml aprotinin, 0.5 mM phenylmethylsulfonyl fluoride, one tablet of Complete proteinase inhibitor [Roche], and 50 units/ml RNase OUT [Invitrogen]). The lysate was incubated on ice for 5 min and centrifuged at 13,000 g for 10 min at 4 °C to pellet cell debris. Clarified lysates were subjected to immunoprecipitation with anti-VIR antibody or normal rabbit IgG (#2729, Cell Signaling Technology, USA) bound to protein A agarose (Sigma-Aldrich). The input RNA and immunoprecipitated RNAs were extracted with TRIzol reagent and reverse-transcribed with the Superscript III First-Strand Synthesis System (Thermo Fisher Scientific, USA)[33,99]. The relative enrichment of each gene was determined by qRT-PCR and calculated by normalizing the amount of a target cDNA fragment against the amount of TUB2 as an internal control, followed by normalizing the value for immunoprecipitated samples against that for the input. The VIR-associated proteins were identified by LC-MS/MS. Volcano plots were generated representing the statistical test results based on the two-sided Student's t tests that were performed comparing the VIR pull-down versus the IgG pull-down, using P-value for truncation as previous methods[100].

## Analysis of mRNA stability

The 7-day-old Col-0 and vir-1 Arabidopsis seedlings grown on half-strength MS medium were transferred to liquid half-strength MS medium containing 0.2 mM actinomycin D[36]. After 30 min of incubation, the seedlings were collected and referred to as 0 h samples, and additional samples were collected at different times. mRNA levels were measured by qRT-PCR. 18S ribosomal RNA was used as the internal reference.

## Reporting summary

Further information on research design is available in the Nature Portfolio Reporting Summary linked to this article.

## Data availability

The MeRIP-seq data have been deposited to the NCBI repository (https://www.ncbi.nlm.nih.gov/sra/PRJNA899541) with the dataset identifier PRJNA899541. The Rio-Seq data have been deposited to the NCBI repository (https://www.ncbi.nlm.nih.gov/sra/PRJNA899535) with the dataset identifier PRJNA899535. The RNA-Seq data have been deposited to the NCBI repository (https://www.ncbi.nlm.nih.gov/sra/PRJNA899318) with the dataset identifier PRJNA899318. The proteomics data (LC-MS/MS for identification of VIR interactors) have been deposited to the ProteomeXchange Consortium (http://proteomecentral.proteomexchange.org/cgi/GetDataset?ID=PXD038080) via the iProX partner repository with the dataset identifier PXD038080. Source data are provided with this paper.

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

## Acknowledgements

The authors are grateful to Prof Rupert Fray for kindly providing the Arabidopsis *mta ABI3prom:MTA* complemented lines. This research was supported by the National Natural Science Foundation of China Grant (31970261, U22A20446), and the Talent Support Project of Guangdong (2019TQ05N182).

## Author contributions

H.-L.J., H.-B.W., and M.Z. conceived the project and designed the study. M.Z. performed most of the experiments and sequencing data analyses. Y.Z., R.P. and J.D. assisted in some RNA extractions and mutant identification. G.L. and Y.L. performed quantification of m⁶A levels in mRNA by LC-MS/MS. K.R. contributed with material (the *vir-1* mutant and its complemented lines, *MTB*-RNAi lines, and *UBQ10prom:XVE»VIR* RNAi constructs). S.D., Z.C., and B.L. assisted in the analysis of some sequencing data. J.R., G.L., and W.Z. provided technical support. M.Z., H.-L.J., and H.-B.W. wrote the manuscript. M.Z., H.-L.J., H.-B.W., K.R., and G.L. revised the manuscript.

## Competing interests

The authors declare no competing interests.
