## [Peer Review File · Nature Communications]

N6 1 -Methyladenosine RNA Modification Regulates Photosynthesis during Photodamage in PlantsReviewer #1 (Remarks to the Author):

N6-Methyladenosilation (m6A) is the most prevalent post-translational mRNA modification in eukaryotes. It has been broadly studied from a developmental perspective in Arabidopsis, and studies are emerging on the role of m6A modifications in the chloroplast. This paper focuses on the role of m6A in the regulation of photosynthesis.

The authors of this paper identified VIR, a writer protein part of the methyltransferase complex, as a crucial regulator of photoprotection. The authors claim that VIR mediates gene expression of many regulatory genes but also key genes in photoprotection via m6A modifications of their mRNA, affecting their abundance and translation.

The influence of VIR in photoprotection is of relevance and has potential applications to the fields of plant RNA and photosynthesis. The fact that m6A modifications affect chloroplast/photosynthetic genes had been shown before (Li et al 2014; Luo et al 2014). In addition, this is not the first publication that looks at the chloroplast epitranscriptome of the vir mutant (Parker et al 2020). However, the novelty of this work lies in its focus on m6A modifications before and after high light stress and the demonstration of the direct interaction between VIR and photoprotection-related genes, while the Parker et al paper focuses on the technicalities of the use of Nanopore sequencing for the detection of m6A modifications at a single-base resolution.

The results are technically sound, and I am confident that their analysis and interpretation have been done carefully. The discussion covers the implications of the key results and other relevant studies. The title and abstract are descriptive of the work and, overall, the claims are supported by sufficient data.

The use of methods including light-response and NPQ measurements, blue native PAGE, and qRT-PCR is justified and appropriately described and interpreted. These were crucial to show the direct effect of VIR mutations in photosynthesis at the physiological, transcriptional and protein level. RNA immunoprecipitation and sequencing of m6A modifications are the predominant approaches for identifying mRNAs with the m6A modification. Methods are described accurately and should ensure reproducibility.

The manuscript is written in a clear and straightforward way, and the data is presented clearly and accurately in the figures.

Minor improvements are suggested below:

1. I believe the manuscript would be strengthened by including previous work on m6A modifications to chloroplast and photosynthetic genes in the introduction. A few sentences after line 96 in page 4 would suffice, and I would argue that it would be important to reference the work by Parker et al on the vir mutant.
2. I would suggest adding statistics to the plots in Figure 2d and 2e to reflect the statistical significance of the differences between the mutant and the controls (similar to Figure 6f).

References

1. Li, Y., Wang, X., Li, C., Hu, S., Yu, J. and Song, S. Transcriptome-wide N6-methyladenosine profiling of rice callus and leaf reveals the presence of tissue-specific competitors involved in selective mRNA modification. *RNA Biol.* 2014, 11, 1180–1188.
2. Luo, G.-Z., MacQueen, A., Zheng, G., Duan, H., Dore, L.C., Lu, Z., Liu, J. et al. Unique features of the m6A methylome in Arabidopsis thaliana. *Nat. Comm.*, 2014, 5, 5630.
3. Parker, M.T.; Knop, K.; Sherwood, A.V.; Schurch, N.J.; Mackinnon, K.; Gould, P.D.; Hall, A.J.W.; Barton, G.J.; Simpson, G.G. Nanopore direct RNA sequencing maps the complexity of Arabidopsis mRNA processing and m6A modification. *Elife* 2020, 9.

Reviewer #2 (Remarks to the Author):

The manuscript by Zhang et al., demonstrated a role of m6A modification in photosynthesis and high light stress in plants. The authors identified a number of photosynthesis-related transcripts with m6A modification and that VIRILIZER (VIR), an m6A methyltransferase complex, is essential for photosynthesis regulation. VIR depletion resulted in lower m6A levels of photosynthesis-related transcripts, leading to reduced transcript stability and translation efficiency. Overall, this study highlights the importance of photosynthesis-related gene regulation at the epitranscriptomic level. Most results and evidence presented in this study generally seem to support the conclusion. However, there are a few problems need to be addressed:

1. The authors found that high light can induce the expression levels of m6A writers as shown in Figure 1e-f. However, the differential expression analysis in Figure 1b showed that more m6A sites are downregulated compared with the upregulated sites. How would the authors explain this discrepancy between them?

2. When the authors performed studies on effect of m6A on transcript level, can the authors test if the transcription or decay are affected by loss of vir? It was not clear. This could be a transcription effect or a combination of transcription and decay. See <https://pubmed.ncbi.nlm.nih.gov/34294912/> and recent mammalian work on roles of m6A on transcription. The NBT paper should be cited as this highlights potential importance of m6A in plant regulation in general.

3. The induction of m6A writers seem to be dramatic after high light treatment (Figure 1e-f). However, the authors did not conduct any experiment to quantify whether this induction of writer genes would lead to the induction of overall transcriptomic m6A level. This is also related to the point above.

4. From lines 147-152, the authors described an experiment measuring the gene expression changes of m6A regulators in response to high light stress. However, the authors did not specify what the "12h recovery" sample is. The figure legend of Figure 1 also did not describe it.

5. From lines 152-155, the authors seemed to conduct a different experiment. However, the description of this experiment is not clear about how it is different from the experiment as described from lines 147-152.

6. It is confusing that the BN-PAGE and CBB panels in Figure 3a look the same with those in Figure S3d. If they are the same images, it is unnecessary to duplicate the same images in the supplementary data simply. The authors can describe this simply by a clear description. Alternatively, different images from the same experiment can be provided to also support the reproducibility.

7. From lines 279-281, the authors concluded that "the loss of VIR causes dramatic changes in the topology of the m6A methylome before and after HL exposure", which is in conflict with the observation that "HL exposure did not alter the overall distribution of m6A in either Col-0 or vir". The authors need to rephrase to clarify.

8. The authors identified vir from their screen. How about MTA and MTB? These might be to essential and mediate both transcription and post-transcription regulation? Perhaps vir affects a portion of mRNA, thus phenotype is less severe? Some discussions will help.

9. The authors observed quite interesting effects of m6A on transcript level. The whole manuscript is descriptive but this is ok. Would be nice to include discussions about potential reader proteins that may read these m6A or other mechanisms that affect transcript level through m6A. This seems to be key for future research. Some speculation and future pull down with the m6A modified RNA probes can be very interesting to ID reader proteins that bind these target transcripts.

Reviewer #3 (Remarks to the Author):

This manuscript reports a role of m6A modification in the regulation of photosynthesis under excess light stress in Arabidopsis. By analyzing the function of VIR, the Arabidopsis m6A writer, on photosynthesis and identifying the photosynthesis-related transcripts regulated by VIR, the manuscript revealed a mechanism for m6A-dependent maintenance of photosynthetic efficiency in plants. Overall, I found this work to contain well-performed experiments but limited novelty. The influence of m6A on photosynthesis-related genes has been mentioned by Luo et al. (Nat Commun, 2014, 5:5630. Unique features of the m6A methylome in Arabidopsis thaliana.), and this was not discussed in this manuscript. Moreover, the role of VIR on m6A modification of photoprotection-related genes or the effect of m6A on mRNA stability and translation efficiency is descriptive. The mechanisms of VIR on m6A modification (e.g. the interacting proteins) and how VIR affects the mRNA destiny were not investigated. As a well-known m6A writer in Arabidopsis, these should be considered.

Specific comments:

1. Line 135-138: Considering that the expression of genes encoding m6A methyltransferase subunits dramatically increased under HL treatment, how to explain why obviously more m6A peaks was downregulated by HL treatment? LC-MS/MS assay should be performed to determine whether HL treatment induces the change of global m6A modification level, i.e. decrease or increase. It will provide valid references for considering the role of other m6A regulators in HL-related photosynthesis.
2. Line 184-189: How to define the "similar developmental stage"? How many seedlings were observed? Concrete descriptions are needed here.
3. Line 409-425: I suggest the authors to transform vir (or vir/hhl1) protoplasts with plasmids overexpressing wild-type or mutant HHL1 to verify whether the effects of HHL1 m6A sites on mRNA stability and translation are VIR-dependent.
4. Figure 2b and 2c: The significance analysis between the VIR-complemented and Col-0/vir is needed. In addition, the "*" should be explained in the figure legend.
5. Figure S3f: For the protein degradation analysis, the degradation curves need to be drawn, depending on the gray values.
6. Figure 7: The working model may lead to misunderstanding and should be adjusted.
 - No clear indication of the working condition (normal or HL treatment) in the photograph, although it was mentioned in the figure legend.
 - No direct evidences supporting that the three photoprotection-related genes could be targeted by the m6A eraser ALKBH10 or readers ECT2/3/4.
 - I might think the RNA stable and translation activity of all three photoprotection-related genes were affected by VIR-mediated m6A modifications according to the arrow indication, but in fact not, according to the results.
 - I suggest the authors to comprehensively investigate the influences of VIR-mediated m6A modifications on gene expression, mRNA stability, and translation efficiency of the three photoprotection-related genes under HL treatment. The relevant content can be appropriately described in the manuscript.
7. Discussion: The authors should discuss why the photochemical efficiency under normal condition in the vir mutant does not change compared with the Col-0, although the m6A levels and protein levels of the concerned photoprotection-related genes significantly decreased without HL treatment.

Responses to Reviewers' Comments:

Thank you very much for your great help to improve the manuscript (MS No. NCOMMS-22-06773). We have conducted additional experiments accordingly and extensively revised the manuscript to fully address your concerns. The following is a point-to-point response to these comments.

For Reviewer #1:

N6-Methyladenosilation (m6A) is the most prevalent post-translational mRNA modification in eukaryotes. It has been broadly studied from a developmental perspective in Arabidopsis, and studies are emerging on the role of m6A modifications in the chloroplast. This paper focuses on the role of m6A in the regulation of photosynthesis. The authors of this paper identified VIR, a writer protein part of the methyltransferase complex, as a crucial regulator of photoprotection. The authors claim that VIR mediates gene expression of many regulatory genes but also key genes in photoprotection via m6A modifications of their mRNA, affecting their abundance and translation. The influence of VIR in photoprotection is of relevance and has potential applications to the fields of plant RNA and photosynthesis. The fact that m6A modifications affect chloroplast/photosynthetic genes had been shown before (Li et al 2014; Luo et al 2014). In addition, this is not the first publication that looks at the chloroplast epitranscriptome of the vir mutant (Parker et al 2020). However, the novelty of this work lies in its focus on m6A modifications before and after high light stress and the demonstration of the direct interaction between VIR and photoprotection-related genes, while the Parker et al paper focuses on the technicalities of the use of Nanopore sequencing for the detection of m6A modifications at a single-base resolution. The results are technically sound, and I am confident that their analysis and interpretation have been done carefully. The discussion covers the implications of the key results and other relevant studies. The title and abstract are descriptive of the work and, overall, the

claims are supported by sufficient data. The use of methods including light-response and NPQ measurements, blue native PAGE, and qRT-PCR is justified and appropriately described and interpreted. These were crucial to show the direct effect of VIR mutations in photosynthesis at the physiological, transcriptional and protein level. RNA immunoprecipitation and sequencing of m6A modifications are the predominant approaches for identifying mRNAs with the m6A modification. Methods are described accurately and should ensure reproducibility. The manuscript is written in a clear and straightforward way, and the data is presented clearly and accurately in the figures.

Response: Thank you very much for your support of this study.

Minor improvements are suggested below:

Comment 1. I believe the manuscript would be strengthened by including previous work on m6A modifications to chloroplast and photosynthetic genes in the introduction. A few sentences after line 96 in page 4 would suffice, and I would argue that it would be important to reference the work by Parker et al on the *vir* mutant.

References

1. Li, Y., Wang, X., Li, C., Hu, S., Yu, J. and Song, S. Transcriptome-wide N6-methyladenosine profiling of rice callus and leaf reveals the presence of tissue-specific competitors involved in selective mRNA modification. *RNA Biol.* 2014, 11, 1180 – 1188.
2. Luo, G.-Z., MacQueen, A., Zheng, G., Duan, H., Dore, L.C., Lu, Z., Liu, J. et al. Unique features of the m6A methylome in *Arabidopsis thaliana*. *Nat. Comm.*, 2014, 5, 5630.
3. Parker, M.T.; Knop, K.; Sherwood, A.V.; Schurch, N.J.; Mackinnon, K.; Gould, P.D.; Hall, A.J.W.; Barton, G.J.; Simpson, G.G. Nanopore direct RNA sequencing maps the complexity of *Arabidopsis* mRNA processing and m6A modification. *Elife* 2020, 9.

Response: Thank you for the constructive comment. Based on your suggestion, we have included previous work on m⁶A modifications of chloroplast and photosynthetic genes (Li et al., 2014; Luo et al., 2014) in the introduction (now Page 4, Line 101-102), and cited the work by Parker et al. on the *vir-1* mutant (Parker et al., 2020) in the discussion of the revised manuscript (now Page 22, Line 587-591).

Comment 2. I would suggest adding statistics to the plots in Figure 2d and 2e to reflect the statistical significance of the differences between the mutant and the controls (similar to Figure 6f).

Response: As requested, we have added the results from statistical analyses to the plots in Figure 2d and 2e in the revised manuscript.

For Reviewer 2#:

The manuscript by Zhang et al., demonstrated a role of m⁶A modification in photosynthesis and high light stress in plants. The authors identified a number of photosynthesis-related transcripts with m⁶A modification and that VIRILIZER (VIR), an m⁶A methyltransferase complex, is essential for photosynthesis regulation. VIR depletion resulted in lower m⁶A levels of photosynthesis-related transcripts, leading to reduced transcript stability and translation efficiency. Overall, this study highlights the importance of photosynthesis-related gene regulation at the epitranscriptomic level. Most results and evidence presented in this study generally seem to support the conclusion. However, there are a few problems need to be addressed:

Response: Thank you for your support and the constructive suggestions.

Comment 1. The authors found that high light can induce the expression levels of m⁶A writers as shown in Figure 1e-f. However, the differential expression analysis in Figure 1b showed that more m⁶A sites are downregulated compared

with the upregulated sites. How would the authors explain this discrepancy between them?

Response: Thank you for the comment. The original Figure 1b (now Figure 1d) showed the number of upregulated and downregulated m⁶A sites, but not the overall abundance difference of m⁶A modification levels. To determine whether HL treatment changes global m⁶A modification levels, we performed dot blot and LC-MS experiments. Both results showed that high light induces an upregulation of overall m⁶A levels in wild-type Arabidopsis plants (now Figure 1a and 1b), which is consistent with the upregulated expression levels of m⁶A writers after high light treatment in Figure 1e (now Figure 1g).

Comment 2. When the authors performed studies on effect of m⁶A on transcript level, can the authors test if the transcription or decay are affected by loss of *vir*? It was not clear. This could be a transcription effect or a combination of transcription and decay. See <https://pubmed.ncbi.nlm.nih.gov/34294912/> and recent mammalian work on roles of m⁶A on transcription. The NBT paper should be cited as this highlights potential importance of m⁶A in plant regulation in general.

Response: Thanks for the constructive comment. Our results showed that the mRNAs of photoprotection-related genes (such as *HHL1* and *MPH1*) were more rapidly degraded in *vir-1* than in Col-0 seedlings (now Figure 6d), indicating that lower levels of the m⁶A modification promotes mRNA decay. Our new data added in the revised manuscript suggested that mRNAs of *HHL1* and *MPH1* containing the RRACH motif in *vir-1* mutants are cleaved more than that in Col-0 plants (now Figure 6e and 6f).

According to the Reviewer's suggestion, we further tested the effect of m⁶A on transcription following the methods of the NBT article. First, we isolated poly(A) RNA from plant tissue samples of equal mass along with a synthesized poly(A) RNA spike-in control and quantified the ratio of poly(A) RNA to spike-in RNA by total RNA Pico Chip analysis using an Agilent 2100 Bioanalyzer. We

determined that *vir-1* mutant plants accumulated slightly higher levels of poly(A) RNA compared to wild-type plants, but this difference was not significant (now Figure S10a). Furthermore, we performed quantitative RNA-seq with an External RNA Controls Consortium (ERCC) RNA spike-in control. We observed that *vir-1* mutant plants and wild-type plants accumulate comparable overall levels of poly(A) RNA (now Figure S10b and S10c), which confirmed that m⁶A does not significantly affect overall transcription. We also cited the NBT paper in the revised manuscript (now Page 14, Line 371-373).

Comment 3. The induction of m⁶A writers seem to be dramatic after high light treatment (Figure 1e-f). However, the authors did not conduct any experiment to quantify whether this induction of writer genes would lead to the induction of overall transcriptomic m⁶A level. This is also related to the point above.

Response: As suggested, we performed dot blot and LC-MS experiments to quantify whether the induction of writer genes increased the overall transcriptomic m⁶A level. Both assays showed that high light induced an upregulation of the overall m⁶A level in wild-type Arabidopsis plants (now Figure 1a and 1b), which is consistent with the upregulated expression levels of m⁶A writers after high light treatment in Figure 1e (now Figure 1g).

Comment 4. From lines 147-152, the authors described an experiment measuring the gene expression changes of m⁶A regulators in response to high light stress. However, the authors did not specify what the “12h recovery” sample is. The figure legend of Figure 1 also did not describe it.

Response: Thank you for the comment. We now specifically describe the “12-h recovery” sample in the figure legend of Figure 1 in the revised manuscript.

Comment 5. From lines 152-155, the authors seemed to conduct a different experiment. However, the description of this experiment is not clear about how it is different from the experiment as described from lines 147-152.

Response: We have rephrased these sentences (now Page 7, Lines 163-169) and clarified the differences between the experiment as described from lines 147-152 (now Page 7-8, Lines 169-178).

Comment 6. It is confusing that the BN-PAGE and CBB panels in Figure 3a look the same with those in Figure S3d. If they are the same images, it is unnecessary to duplicate the same images in the supplementary data simply. The authors can describe this simply by a clear description. Alternatively, different images from the same experiment can be provided to also support the reproducibility.

Response: The BN-PAGE and CBB panels in Figure 3a were indeed the same as those in Figure S3d. According to the Reviewer's suggestion, we have deleted the image in Figure S3d, and clarified this in the legend of Figure S4d in the revised manuscript.

Comment 7. From lines 279-281, the authors concluded that “the loss of VIR causes dramatic changes in the topology of the m⁶A methylome before and after HL exposure”, which is in conflict with the observation that “HL exposure did not alter the overall distribution of m⁶A in either Col-0 or vir”. The authors need to rephrase to clarify.

Response: We meant that the topology of the m⁶A methylome in the *vir-1* mutant showed dramatic changes compared to that in the wild type, regardless of HL treatment. However, the overall distribution of m⁶A after HL treatment was not altered compared to before HL treatment, in both Col-0 and *vir-1*. We have rephrased and clarified this (now Page 12, Lines 309-312).

Comment 8. The authors identified *vir* from their screen. How about MTA and MTB? These might be too essential and mediate both transcription and post-transcription regulation? Perhaps *vir* affects a portion of mRNA, thus phenotype is less severe? Some discussions will help.

Response: Thank you for the constructive comment. We detected the F_v/F_m values of *ABI3:MTA* and *MTB-RNAi* lines after HL treatment. We observed that the F_v/F_m values of *ABI3:MTA* were lower than control values, but there was no significant difference between *MTB-RNAi* and the control (now Figure S3). These results suggest that *vir-1* may affect a portion of mRNA, and other writers also contribute to the phenotype. We have discussed this in the Discussion (now Page 22, Lines 581-586).

Comment 9. The authors observed quite interesting effects of m6A on transcript level. The whole manuscript is descriptive but this is ok. Would be nice to include discussions about potential reader proteins that may read these m6A or other mechanisms that affect transcript level through m6A. This seems to be key for future research. Some speculation and future pull down with the m6A modified RNA probes can be very interesting to ID reader proteins that bind these target transcripts.

Response: Thanks for the constructive comment. According to the Reviewer's suggestion, we discussed this in the Discussion of the revised manuscript (now Page 25, Lines 671-674).

For Reviewer #3:

This manuscript reports a role of m6A modification in the regulation of photosynthesis under excess light stress in Arabidopsis. By analyzing the function of VIR, the Arabidopsis m6A writer, on photosynthesis and identifying the photosynthesis-related transcripts regulated by VIR, the manuscript revealed a mechanism for m6A-dependent maintenance of photosynthetic efficiency in plants. Overall, I found this work to contain well-performed experiments but limited novelty. The influence of m6A on photosynthesis-related genes has been mentioned by Luo et al. (Nat Commun, 2014, 5:5630. Unique features of the m6A methylome in Arabidopsis thaliana.), and this was

not discussed in this manuscript. Moreover, the role of VIR on m⁶A modification of photoprotection-related genes or the effect of m⁶A on mRNA stability and translation efficiency is descriptive. The mechanisms of VIR on m⁶A modification (e.g. the interacting proteins) and how VIR affects the mRNA destiny were not investigated. As a well-known m⁶A writer in Arabidopsis, these should be considered.

Response: Thank you for the constructive comment. Luo et al. deal with some transcriptomics but didn't show much about functional relevance of photosynthesis (Luo et al., 2014). The functional confirmation of role of m⁶A in photosynthesis, which has not been for 8 years unknown. The novelty of this work lies in its focus on m⁶A modifications before and after HL stress and the demonstration of a direct interaction between VIR and photoprotection-related genes, as commented by Reviewer #1. This study also provided the putative mechanistic explanation of role of VIR/m⁶A in photoprotection of methylated transcripts for maintenance of efficient photosynthesis during photodamage.

We also cited the 2014 paper (Nat Commun, 2014, 5:5630. Unique features of the m⁶A methylome in Arabidopsis thaliana.) and discussed it in the Discussion of the revised manuscript (now Page 20, Lines 540-542).

We further explored the mechanisms of VIR on m⁶A modification (e.g., the interacting proteins) and how VIR affects mRNA fate.

First, we performed RNA immunoprecipitation (RIP) assays (now Figure 7a), and identified VIR-associated proteins by LC-MS/MS analysis. The results indicate that VIR interacts with a large number of regulators related to RNA stability, processing and translation (now Figure 7e and 7f), suggesting the general role of VIR in post-transcriptional regulation. Intriguingly, another m⁶A writher, HAKAI (Růžička et al., 2017), interacted with VIR (Figure 7f), supporting the idea that the m⁶A regulators may interact in m⁶A modification of photoprotection-related transcripts.

m⁶A stabilizes mRNAs by inhibiting local ribonucleolytic cleavage (Anderson et al., 2018). We thus explored whether the lower mRNA density

and faster degradation in *vir-1* result from local ribonucleolytic cleavage. Based on the predicted motifs of the highest-cleaved nucleotide within m⁶A peaks (Anderson et al., 2018), we compared the cleavage levels around motifs of the highest-cleaved nucleotide within different-level m⁶A peaks in *vir-1* and Col-0 plants by qPCR. The relative amounts of *HHL1* and *MPH1* mRNA containing the cleavage sites (RRACH motif) were decreased by 22% and 17%, respectively, in *vir-1* compared to Col-0 (now Figure 6e and 6f), suggesting that *HHL1* and *MPH1* mRNA containing the RRACH motif are cleaved more in *vir-1* mutants than in Col-0 plants.

The possible working model of VIR-dependent m⁶A participation in photoprotection in plants may be that, in wild-type Arabidopsis, VIR associates with photoprotection-related mRNAs and modify their m⁶A modifications, which recruit mRNA regulators, which promotes their stability (such as *HHL1* and *MPH1*), and translation efficiency (such as *STN8*) and thus improves photoprotection capacity and photosynthetic efficiency under excess light conditions. In *vir-1* mutants, VIR deficiency results in defects of m⁶A modifications of photoprotection-related mRNA, which fail to associate with mRNA regulators, and decreases their stability due to ribonucleolytic cleavage (such as *HHL1* and *MPH1*) and translation efficiency (such as *STN8*), and reduces photoprotection capacity and photosynthetic efficiency under excess light.

Specific comments:

1. Line 135-138: Considering that the expression of genes encoding m⁶A methyltransferase subunits dramatically increased under HL treatment, how to explain why obviously more m⁶A peaks was downregulated by HL treatment? LC-MS/MS assay should be performed to determine whether HL treatment induces the change of global m⁶A modification level, i.e. decrease or increase.

It will provide valid references for considering the role of other m⁶A regulators in HL-related photosynthesis.

Response: Thanks for the comment. Figure 1b showed the number of upregulated and downregulated m⁶A sites, but not the difference in abundance of the m⁶A modification. According to the Reviewer's suggestion, we performed dot blot and LC-MS experiments to determine whether HL treatment alters the global m⁶A modification level. Both assays showed that HL induced an upregulation of the overall m⁶A level in wild-type Arabidopsis plants (now Figure 1a and 1b), which is consistent with the upregulated expression levels of m⁶A writers after HL treatment in Figure 1e (now Figure 1g).

2. Line 184-189: How to define the "similar developmental stage"? How many seedlings were observed? Concrete descriptions are needed here.

Response: Although the *vir-1* mutant and wild-type plants were the same age, most mutant plants were less developed, and some *vir-1* mutant plants varied in size. Therefore, we selected mutant and Col-0 plants at a similar developmental stage (similar sized plants). More than 100 seedlings were observed in our experiments. We have added this text to the revised manuscript (now Page 9 Lines 207-212). Moreover, we corroborated the observed different photosynthetic rates in the *vir* hypomorph including the *VIR*-RNAi data in the revised manuscript.

3. Line 409-425: I suggest the authors to transform *vir* (or *vir/hhl1*) protoplasts with plasmids overexpressing wild-type or mutant HHL1 to verify whether the effects of HHL1 m⁶A sites on mRNA stability and translation are VIR-dependent.

Response: According to the Reviewer's suggestion, we transformed *vir-1* protoplasts with plasmids overexpressing wild-type or mutant *HHL1* to determine if the effects of *HHL1* m⁶A sites on protein abundance are VIR-dependent. The results showed that HHL1 abundance only decreased to ~67% of wild-type levels in *vir-1* protoplasts expressing A641,661,670-G mutant

construct (now Figure S18), but decreased to ~25% of wild-type levels in *hhl1* protoplasts expressing A641,661,670-G mutants (now Figure 6j). This suggests that the effect of *HHL1* m⁶A sites on protein abundance are VIR-dependent.

4. Figure 2b and 2c: The significance analysis between the VIR-complemented and Col-0/*vir* is needed. In addition, the “*” should be explained in the figure legend.

Response: According to the Reviewer’s suggestion, we performed a statistical analysis between the VIR-complemented and Col-0/*vir-1*. We also explained the “*” in the figure legend (now Figure 2b and 2c). Values are means ± SE of three biological replicates. *, $P < 0.05$; **, $P < 0.01$, by Student’s *t*-test.

5. Figure S3f: For the protein degradation analysis, the degradation curves need to be drawn, depending on the gray values.

Response: According to the Reviewer’s suggestion, we quantified the band intensity in all immunoblots for the protein degradation analysis (now Figure S7b). The degradation curves further confirmed no significant differences for protein degradation in wild type and *vir-1* mutant plants.

6. Figure 7: The working model may lead to misunderstanding and should be adjusted.

-No clear indication of the working condition (normal or HL treatment) in the photograph, although it was mentioned in the figure legend.

-No direct evidences supporting that the three photoprotection-related genes could be targeted by the m⁶A eraser ALKBH10 or readers ECT2/3/4.

-I might think the RNA stable and translation activity of all three photoprotection-related genes were affected by VIR-mediated m⁶A modifications according to the arrow indication, but in fact not, according to the results.

-I suggest the authors to comprehensively investigate the influences of VIR-mediated m⁶A modifications on gene expression, mRNA stability, and

translation efficiency of the three photoprotection-related genes under HL treatment. The relevant content can be appropriately described in the manuscript.

Response: Thank you for the constructive suggestion. We adjusted the working model (now Figure 7g) according to the Reviewer's suggestion and our new data.

7. Discussion: The authors should discuss why the photochemical efficiency under normal condition in the *vir* mutant does not change compared with the Col-0, although the m⁶A levels and protein levels of the concerned photoprotection-related genes significantly decreased without HL treatment.

Response: According to the Reviewer's suggestion, we now discuss why the photochemical efficiency does not change under normal conditions in the *vir-1* mutant compared to Col-0, although the m⁶A levels and protein levels of the photoprotection-related genes significantly decreased without HL treatment. This is because photoprotection factors mainly function in photodamage from HL stress, which is similar to that of the mutants of photoprotection-related genes, such as *HHL1* (Jin et al., 2014) or *MPH1* (Liu and Last, 2015) (now Page 23, Lines 604-609).

References

- Anderson, S.J., Kramer, M.C., Gosai, S.J., Yu, X., Vandivier, L.E., Nelson, A.D., Anderson, Z.D., Beilstein, M.A., Fray, R.G., and Lyons, E.** (2018). N⁶-methyladenosine inhibits local ribonucleolytic cleavage to stabilize mRNAs in *Arabidopsis*. *Cell reports* **25**, 1146-1157. e1143.
- Jin, H., Liu, B., Luo, L., Feng, D., Wang, P., Liu, J., Da, Q., He, Y., Qi, K., and Wang, J.** (2014). HYPERSENSITIVE TO HIGH LIGHT1 interacts with LOW QUANTUM YIELD OF PHOTOSYSTEM II1 and functions in protection of photosystem II from photodamage in *Arabidopsis*. *The Plant Cell* **26**, 1213-1229.
- Li, Y., Wang, X., Li, C., Hu, S., Yu, J., and Song, S.** (2014). Transcriptome-wide N⁶-methyladenosine profiling of rice callus and leaf reveals the presence of tissue-specific competitors involved in selective mRNA modification. *RNA Biology* **11**, 1180-1188.
- Liu, J., and Last, R.L.** (2015). A land plant - specific thylakoid membrane protein contributes to photosystem II maintenance in *A. thaliana*. *The Plant Journal* **82**, 731-743.
- Luo, G.-Z., MacQueen, A., Zheng, G., Duan, H., Dore, L.C., Lu, Z., Liu, J., Chen, K., Jia, G., and Bergelson, J.** (2014). Unique features of the m⁶A methylome in *Arabidopsis thaliana*. *Nature communications* **5**, 1-8.
- Parker, M.T., Knop, K., Sherwood, A.V., Schurch, N.J., Mackinnon, K., Gould, P.D., Hall, A.J., Barton, G.J., and Simpson, G.G.** (2020). Nanopore direct RNA sequencing maps the complexity of *Arabidopsis* mRNA processing and m⁶A modification. *Elife* **9**, e49658.
- Růžička, K., Zhang, M., Campilho, A., Bodi, Z., Kashif, M., Saleh, M., Eeckhout, D., El - Showk, S., Li, H., and Zhong, S.** (2017). Identification of factors required for m⁶A mRNA methylation in *Arabidopsis* reveals a role for the conserved E3 ubiquitin ligase HAKAI. *New Phytol.* **215**, 157-172.

Reviewer #1 (Remarks to the Author):

Thank you for submitting a revised version of the manuscript. My comments have been addressed appropriately and I believe the manuscript has been enhanced with the additional experiments and edits.

Reviewer #2 (Remarks to the Author):

The authors have addressed my comments. I do have a suggestion. The authors used mass spec and Dot Blot to measure the overall m6A level change with or without high light. I wonder if they can integrate m6A peaks from their MeRIP-seq results to reach the same conclusion.

Reviewer #3 (Remarks to the Author):

The authors have addressed all my concerns and improved the manuscript substantially.

Responses to Reviewers' Comments:

Thank you very much for your great help to improve the manuscript (MS No. NCOMMS-22-06773A). The following is a point-to-point response to the comments.

For Reviewer 2#:

The authors have addressed my comments. I do have a suggestion. The authors used mass spec and Dot Blot to measure the overall m⁶A level change with or without high light. I wonder if they can integrate m⁶A peaks from their MeRIP-seq results to reach the same conclusion.

Response: Thank you for your support and the constructive suggestions. MeRIP-seq is not suited for quantification of modified nucleosides due to that recognition by antibodies is influenced by the RNA sequence context and this may introduce bias by detecting certain methylation sites to the detriment of others, although it has advantages in modification mapping and generation of a large number of data points for each RNA species (Molinie et al., 2016; Hartstock and Rentmeister, 2019; Zheng et al., 2020). To date, the MS approaches, especially LC-MS/MS, are particularly suited for quantifying modified nucleosides (Thüring et al., 2017; Amalric et al., 2022). Thus, we did not integrate m⁶A peaks from the MeRIP-seq to quantify the overall m⁶A level.

References

- Amalric, A., Bastide, A., Attina, A., Choquet, A., Vialaret, J., Lehmann, S., David, A., and Hirtz, C.** (2022). Quantifying RNA modifications by mass spectrometry: a novel source of biomarkers in oncology. *Critical Reviews in Clinical Laboratory Sciences* **59**, 1-18.
- Hartstock, K., and Rentmeister, A.** (2019). Mapping N6 - Methyladenosine (m6A) in RNA: Established Methods, Remaining Challenges, and Emerging Approaches. *Chemistry—A European Journal* **25**, 3455-3464.
- Molinie, B., Wang, J., Lim, K.S., Hillebrand, R., Lu, Z.-x., Van Wittenberghe, N., Howard, B.D., Daneshvar, K., Mullen, A.C., and Dedon, P.** (2016). m6A-LAIC-seq reveals the census and complexity of the m6A epitranscriptome. *Nat. Methods* **13**, 692-698.
- Thüring, K., Schmid, K., Keller, P., and Helm, M.** (2017). LC-MS analysis of methylated RNA. In *RNA Methylation* (Springer), pp. 3-18.
- Zheng, H.-x., Zhang, X.-s., and Sui, N.** (2020). Advances in the profiling of N6-methyladenosine (m6A) modifications. *Biotechnol. Adv.* **45**, 107656.